



# Effects of boundary conditions and aquifer parameters on salinity distribution and mixing controlled reactions in high-energy beach aquifers

Rena Meyer[1*], Janek Greskowiak[1], Stephan L. Seibert[1], Vincent E. Post[2], Gudrun Massmann[1]

[1]Institute for Biology and Environmental Sciences, School of Mathematics and Science, Carl von Ossietzky Universität Oldenburg, Ammerländer Heerstraße 114-118, 26129 Oldenburg, Germany

[2]Edinsi Groundwater, Nederhorst den Berg, the Netherlands

*Correspondence to*: Rena Meyer (rena.meyer@uol.de)

**Abstract.**

In high-energy beach aquifers fresh groundwater mixes with recirculating saltwater and biogeochemical reactions modify the composition of groundwater discharging to the sea. Changing beach morphology, hydrodynamic forces as well as hydrogeological properties control density-driven groundwater flow and transport processes that affect the distribution of chemical reactants. In the present study, density-driven flow and transport modelling of a generic 2-D cross-shore transect was

conducted. Boundary conditions and aquifer parameters were varied in a systematic manner in a suite of twenty-four cases. The objective was to investigate their individual effects on flow regime, salt distribution, and potential for mixing controlled chemical reactions in a system with a temporally-variable beach morphology. Our results show that a changing beach morphology causes the migration of infiltration and exfiltration locations along the beach transect that lead to transient flow

and salt transport patterns in the subsurface, thereby enhancing mixing controlled reactions. The shape and extent of the zone where mixing controlled reactions potentially take place as well as the spatio-temporal variability of the freshwater-saltwater interfaces are most sensitive to variable beach morphology, storm floods, hydraulic conductivity and dispersivity.



## 1 Introduction

Sandy beaches make up about 30% of the world's coastline (Luijendijk et al., 2018) and form the transition zone between the terrestrial and marine environment. The subsurface part of this transition zone is called subterranean estuary (STE) (Moore, 1999; Robinson et al., 2006). Here, two water bodies, terrestrial fresh groundwater and recirculating sea water, distinct in physical properties and chemical composition (e.g. density, temperature, pH, redox state) mix and (bio)geochemical reactions take place that change the composition of the water (Anschutz et al., 2009). These reactions and resulting element fluxes across

the land-sea interface are linked to residence times and dispersive mixing processes that depend on dynamic density-driven groundwater flow and transport processes (Anwar et al., 2014; Robinson et al., 2009; Spiteri et al., 2008).

The established concept of flow and transport patterns in the STE describes the relatively stable formation of three water bodies: The freshwater discharge tube (FDT) separates the wave and tide induced upper saline plume (USP) from the saltwater wedge (SW) (Robinson et al., 2006, 2018). At the interfaces of the waterbodies dispersive mixing zones develop. The extent

of the mixing zone where (bio)geochemical reactions take place depends on the geological conditions and (hydro-)dynamic forces (Michael et al., 2016; Robinson et al., 2018). In a homogenous aquifer, three distinct Darcy velocity regimes exist: (1) very low velocities in the SW, (2) high velocities in the discharge zone of the FDT and (3) uniform velocities in the coastal aquifer on the landward side (Costall et al., 2020). The geological architecture and resulting hydrogeological parameters define the connection of the land sea interface and affect the development and extension of the mixing zone where (bio)geochemical

reactions take place. Not only larger geological structures, like paleo-channels (Meyer et al., 2018b, 2019; Mulligan et al., 2007) but also small scale aquifer heterogeneity coupled with tides (Geng et al., 2020) create more complex flow paths for both intruding seawater (SWI) and discharging groundwater rich in nutrients. Costall et al. (2020) demonstrated the effect of heterogeneous hydraulic conductivity on the near shore flow regime resulting in highly variable flow directions and velocities. Michael et al. (2016) showed that geologic complexity in terms of heterogeneous hydraulic conductivity and anisotropy

enhances mixing and focusses the flow across the land-sea interface. Abarca et al. (2007) investigated the effect of longitudinal $\alpha_L$ and transversal $\alpha_T$ dispersivity on the thickness of the mixing zone and found that both contribute to the same extent but each in a different part of the mixing zone.





The impact of hydrodynamic forces such as tides, waves, storm surges on flow regimes and salinity pattern in the STE environment have been comprehensively studied (e.g. Michael et al. (2016), Robinson et al. (2018)). However, most of these

studies focused on low energy environments like lagoons (Müller et al., 2021) or bays while high energy beaches, with high tidal amplitudes and/or high waves were studied less, albeit they are particularly affected by tides, waves and storms (Massmann et al., 2023). Hydraulic and atmospheric forces also continuously re-shape the beach morphology by erosion and accretion processes (Short and Jackson, 2013) that in turn change the hydraulic gradients in the beach subsurface and hence effect the flow and transport regime. It has been recognized in the STE research that the beach topography has an impact on

the subsurface salinity distribution (Abarca et al., 2013; Grünenbaum et al., 2020b; Robinson et al., 2006). These studies found field evidence for the (at least temporal) occurrence of more than one USP and modelled STEs with different, however steady beach slopes (Abarca et al., 2013), or typical sandy beach surfaces representing runnel-ridge (Grünenbaum et al., 2020b) or through-berm systems (Robinson et al., 2006) in comparison to a linearly sloping beach. Robinson et al. (2006) concluded that the beach morphology combined with a tidal signal significantly affects the flow and recirculation of seawater in the STE and

hypothesized that combined with waves this would enhance the exchange of water across the land-sea interface and would led to more complex flow patterns. However, none of the aforementioned studies implemented a transient, continuously changing, beach morphology in their groundwater models. Using a density-dependent groundwater flow and transport model Greskowiak and Massmann (2021) demonstrated that a transient beach morphology combined with seasonal storm floods led to strong spatio-temporal variability in groundwater flow and transport patterns. In their model, the effects of the morphodynamics

reached tens of meters into the subsurface and distorted the typical salinity stratification in the STE. In a next step, Greskowiak et al. (2023) extended the approach by a numerical reactive transport model and analysed the development of redox zones. They concluded that redox zone dynamics in the STE are strongly affected by beach morphodynamics. While some redox reactions take place in the USPs and storm flood affected area, mixing controlled reactions driven by mixing of two solutes in different end members Heiss et al. (2017) occur in the fringes of the USP and at the SW interface. Real world examples relevant

in STE environments for these mixing controlled reactions are iron curtain formation (Charette and Sholkovitz, 2002) or nitrification (Ullman et al., 2003). Anwar et al. (2014) studied the impact of tides and waves on the mixing dependent reactions and concluded that they intensify seawater-freshwater mixing and nutrient transformations, resulting in enhanced fluxes to the

sea. In addition, Heiss et al. (2017) found that tidal amplitude and hydraulic conductivity are the factors that have the strongest effect on nitrate transformation and that the size of the mixing zone as well as solute supply defines the mixing dependent

reactivity.

In their review on geochemical fluxes in sandy beaches Geng et al. (2021) disclosed the need for coupling shoreline morphology and groundwater flow. As Greskowiak and Massmann (2021) showed this is particularly relevant in the understudied environment of high energy beaches, since these are exposed to strong hydrodynamic and atmospheric forces (Massmann et al., 2023) and hence experience profound changes in beach morphology (Montaño et al., 2020). This in turn

shows significant effects on the flow and transport regime reaching deep in the subsurface and impacts the mixing controlled reactions and hence the exchange and transformation of nutrients across the land-sea transition zone.

The objective of the present study is to investigate the interplay of morphological changes and hydrodynamic boundary conditions paired with aquifer properties in the subsurface of high energy beaches in a 2-D density-dependent generic modeling approach. Conditions are based on those of a meso-tidal high-energy beach on Spiekeroog Island in the North Sea. Specifically,

the aims are to systematically evaluate the effect of hydrogeological parameters, i.e. horizontal ($K_h$) and vertical ($K_v$) hydraulic conductivities, , longitudinal ($\alpha_L$) and transversal ($\alpha_T$) dispersivities, porosity (n) and specific storage (spec. stor.), as well as boundary conditions (freshwater inflow, recharge, storm surges) combined with varying beach morphology on (1) the flow regime, (2) the distribution of total dissolved solids (TDS) and their variability, and (3) the potential extent at which mixing controlled chemical reactions occur, hereafter called 'mixing controlled reaction potential'. By closing this research gap, the

present study advances the understanding of subsurface processes in STEs underneath high energy beaches that alter the composition of water discharging into the sea.

## 2 Material and Methods

We used a 2-D generic modelling approach loosely based on real field conditions of the east Frisian barrier island Spiekeroog located in the Wadden Sea in the Southern German Bight. Spiekeroogs' north-facing beach is characterized by meso-tidal

conditions(Hayes, 1979) with a mean tidal range of 2.7 m and mean significant wave height of 1.4 m (Herrling and Winter, 2015). The mean high water line (MHWL) is located at 1.35 m and the mean low water line (MLWL) at -1.35 m (Massmann





et al., 2023). The island receives groundwater recharge of 350-400 mm/a which is about 50% of the mean annual precipitation

of approx. 800 mm (Röper et al., 2012). The upper aquifer below the beach consists of beach, tidal flat and glacial sediments

from Holocene and Pleistocene origin (Massmann et al., 2023; Streif, 1990) and is bounded at the base by an aquitard defined

by a supposedly continuous clay layer at a depth of approx. 40 m (Röper et al., 2012). The model of Greskowiak and Massmann

(Greskowiak and Massmann, 2021) in a slightly adapted version serves as a base case for the 24 simulation cases in this study.

The set-up of the base case is briefly resumed below, while the 24 simulation cases with their respective changes to the base

case are presented in Table 1.

## 2.1 Model approach

Density driven flow and transport were simulated with SEAWAT4 (Langevin et al., 2007), which couples the modules of

groundwater flow (MODFLOW 2000 (Harbaugh et al., 2000)) and multispecies transport (MT3DMS (Zheng and Wang,

1999)). Model input and output were processed with the Python package FloPy (Bakker et al., 2016; USGS, 2021). To

investigate the mixing controlled reaction potential, a simple reaction model was implemented with PHT3D (Prommer and

Post, 2010) that couples SEAWAT with PHREEQC (Parkhurst and Appelo, 1999).

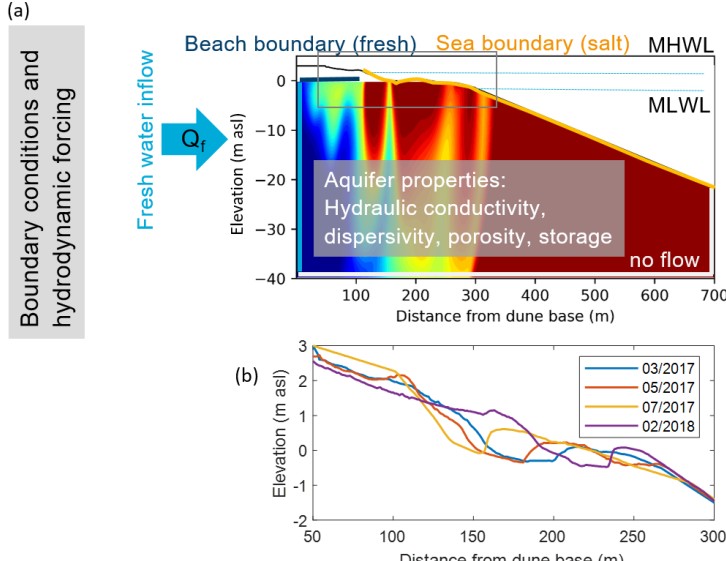

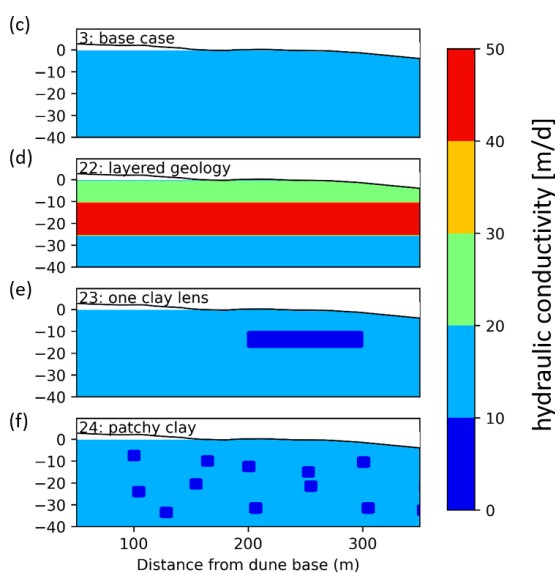






**Figure 1: (a) Model set up with dimensions, boundary conditions and parameters, (colors indicate salt distribution: red = salty, blue = fresh). (b) Four LIDAR scans(Grünenbaum et al., 2020a) of intertidal topography used for interpolation of morphological changes. (c) – (f) Four different geological settings, note K = 0.005 m/d for clay lens and clay patches (dark blue).**


The transient 2-D model representing a 700m long cross-shore transect was discretized into 350 columns of 2 m each and 80 layers with a thickness of 0.5 m each, except for the first layer. The top of the first layer was set to 3 m asl at the upper beach and -0.1 m asl in the intertidal zone to avoid re-wetting problems. The bottom of the first layer was set to -0.5 m asl for all cells. The 2-D approach was justified since the groundwater flow is directed predominantly from the islands' freshwater lens 120 towards the shoreline. The simulation time was 20 years with daily time steps. No flow boundaries (Neumann type) were defined at the northern vertical sea boundary and at the aquifer base. Freshwater (salt concentration = 0) entering from the islands' inland (freshwater lens) along the vertical Southern boundary was prescribed using a specified flux that was uniformly distributed across the cells of the first column (Fig. 1). Meteoric groundwater recharge was applied at the upper beach above the MHWL. A general head boundary (GHB) was specified along the seaside and in the intertidal zone (Fig. 1) with a high 125 conductance of 1000 m²/d to ensure a good connection to the aquifer. The hydraulic head of the GHB boundary was set to 0 below the MLWL. Above the MLWL the tide-averaged approach (Vandenbohede and Lebbe, 2007) was applied and desaturation was accounted for by reducing the head by 17 cm (Greskowiak and Massmann, 2021) at the HWL and linearly decreasing to 0 cm at the MWL (Greskowiak and Massmann, 2021). Based on the approach by Greskowiak and Massmann (2021), daily topography changes in the intertidal zone were linearly interpolated from four lidar scan profiles (Fig1, b) 130 (Greskowiak and Massmann, 2021; Grünenbaum et al., 2020a). Three storm surges were implemented by directly applying seawater recharge above the high tide mark for one day in each winter month, whereby water fluxes were calculated based on the thickness of the unsaturated zone multiplied by the porosity (Holt et al., 2019). Saltwater entered the model domain via a non-dispersive flux boundary and the simulated salinity was assigned to water discharging across the ocean boundary. The initial salt distribution was fresh (0 g/l) landwards from the MLWL and saline seawards of it. Aquifer and boundary conditions 135 are listed in Table 1.





### 2.1.1 Mixing controlled reaction potential

Instead of modelling real mixing reactions, such as iron oxide precipitation at oxic-anoxic interfaces (Charette and Sholkovitz, 2002), we adapted the approach by Perez et al. (2023) and Valocchi et al. (2019) who used a hypothetical reaction potential where a solute C is produced by mixing of solutes A and B in different groundwater end-members. Specifically, we considered a theoretical mixing scenario where two mobile reactants Rf and Rs, entered the system via the freshwater and saltwater boundaries respectively and were transported with the groundwater through the STE. When they mixed, they produced the immobile mixing reaction product Mp via the following reaction rate formulation, which was implemented in the model using PHT3D Eq. 1:

$$r_{Mp} = d[Mp]/dt = k * [Rf]*[Rs] \hspace{2cm} [1]$$

where [Rs] is the concentration of the saltwater reactant Rs, [Rf] is the concentration of the freshwater reactant Rf; [Mp] is the concentration of the immobile mixing product Mp, k is the reaction rate constant and $r_{Mp}$ is the formation rate of Mp. Note that Rf and Rs were not removed by this process and the formed Mp accumulated during the simulation time of 20 years. Hence, at the end of each simulation, Mp was a measure for the extent and intensity of the mixing controlled reaction potential. Note also that the reaction rate constant was arbitrarily set to $k = 1e-7$ $(mol/L)^{-1}$ $s^{-1}$ for all simulation cases. As Rf and Rs were not removed by this processes the value of k has no further meaning, as with that the relative differences of the mixing controlled reaction potential between the different simulation cases are independent from the value of k. The reaction network was implemented into the PHT3D modelling framework. The initial distribution of Rf and Rs, each with a concentration of 1e-3 (mol/L), was in line with the initial salt distribution with Rf present landwards from the MLWL and Rs seawards of it.

### 2.2 Model cases and case evaluation

Twenty-four different model cases were systematically tested covering six aquifer parameters ($K_h$, $K_v$, $\alpha_L$, $\alpha_T$, n, spec. stor.) and three boundary conditions (beach recharge, freshwater inflow from the island, number of storm floods). Each parameter was changed to significantly higher and lower values compared to the base case while still being in a realistic range for sandy STEs in a temperate climate (Table 1). Furthermore, cases with a stable beach morphology with and without storm floods as





well as with three different permeability distributions (Fig. 1, c-f) were tested. All parameters and boundary conditions are

listed in Table 1.

**Table 1: Aquifer properties and boundary conditions for the 24 model cases. Note that the base case is a case with a dynamic topography resembling the model by Greskowiak and Massmann (2021).**

| Case number | $K_h$ [m/d] | $K_v$ [m/d] | $\alpha_L$ [m] | $\alpha_T/\alpha_L$ | n [-] | Spec. stor. [1/m] | $Q_f$ [m3/d/m] | Storm floods per year/ days between storm floods | Description | Cluster |
|---|---|---|---|---|---|---|---|---|---|---|
| 1 | 11 | 5.5 | 2 | 0.1 | 0.35 | 1e-5 | 0.5 | 0/30 | Stable case | B |
| 2 | 11 | 5.5 | 2 | 0.1 | 0.35 | 1e-5 | 0.5 | 3/30 | Stable with SF | B |
| 3 | 11 | 5.5 | 2 | 0.1 | 0.35 | 1e-5 | 0.5 | 3/30 | base case | A |
| 4 | **110** | 5.5 | -"- | 0.1 | 0.35 | 1e-5 | 0.5 | 3/30 | Higher K | C |
| 5 | **1.1** | 0.55 | -"- | 0.1 | 0.35 | 1e-5 | 0.5 | 3/30 | Lower K | B |
| 6 | 11 | **1.1** | 2 | 0.1 | 0.35 | 1e-5 | 0.5 | 3/30 | Higher K anisotropy | C |
| 7 | 11 | **9.9** | 2 | 0.1 | 0.35 | 1e-5 | 0.5 | 3/30 | Lower K anisotropy | A |
| 8 | 11 | 5.5 | **20** | 0.1 | 0.35 | 1e-5 | 0.5 | 3/30 | Higher $\alpha_L$ | |
| 9 | 11 | 5.5 | **0.2** | 0.1 | 0.35 | 1e-5 | 0.5 | 3/30 | Lower $\alpha_L$ | A |
| 10 | 11 | 5.5 | 2 | **0.05** | 0.35 | 1e-5 | 0.5 | 3/30 | higher anisotropy ($\alpha_T/\alpha_L$) | A |
| 11 | 11 | 5.5 | 2 | **0.5** | 0.35 | 1e-5 | 0.5 | 3/30 | Lower anisotropy ($\alpha_T/\alpha_L$) | C |
| 12 | 11 | 5.5 | 2 | 0.1 | 0.35 | **1e-4** | 0.5 | 3/30 | Higher Spec. stor. | A |
| 13 | 11 | 5.5 | 2 | 0.1 | 0.35 | **1e-6** | 0.5 | 3/30 | Lower Spec. stor. | A |
| 14 | 11 | 5.5 | 2 | 0.1 | **0.5** | 1e-5 | 0.5 | 3/30 | n = 0.5 | A |
| 15 | 11 | 5.5 | 2 | 0.1 | **0.2** | 1e-5 | 0.5 | 3/30 | n=0.2 | A |
| 16 | 11 | 5.5 | 2 | 0.1 | 0.35 | 1e-5 | **0.8** | 3/30 | Enhanced $Q_f$ | A |
| 17 | 11 | 5.5 | 2 | 0.1 | 0.35 | 1e-5 | **0.2** | 3/30 | Reduced $Q_f$ | C |
| 18 | 11 | 5.5 | 2 | 0.1 | 0.35 | 1e-5 | 0.5 | 3/30 | Enhanced beach recharge (600 mm) | A |
| 19 | 11 | 5.5 | 2 | 0.1 | 0.35 | 1e-5 | 0.5 | 3/30 | Reduced beach recharge (200 mm) | A |





| | | | | | | | | | | |
|---|---|---|---|---|---|---|---|---|---|---|
| **20** | 11 | 5.5 | 2 | 0.1 | 0.35 | 1e-5 | 0.5 | **6/15** | Six floods | A |
| **21** | 11 | 5.5 | 2 | 0.1 | 0.35 | 1e-5 | 0.5 | **0/30** | No storm flood | B |
| **22** | **Layered (20/50/25)** | 0.5 | 2 | 0.1 | 0.35 | 1e-5 | 0.5 | 3/30 | Layered geology | |
| **23** | **11 + clay lens (0.005)** | 5.5 | 2 | 0.1 | 0.35 | 1e-5 | 0.5 | 3/30 | one clay lens | A |
| **24** | **11 + patchy clay (0.005)** | 5.5 | 2 | 0.1 | 0.35 | 1e-5 | 0.5 | 3/30 | patchy clay | A |






The model results were evaluated according to (1) the flow regime visualized as flow lines (Fig. 2, Fig. 3); (2) the TDS distribution shown as snapshots at the end of the simulation (Fig.. 3) as well as the standard deviation of the TDS concentration (SD) in each cell over the last 10 years of simulation time (Fig.. 4); and (3) the reaction potential ($RP_c$ = sum of accumulated mixing products in each cell ($Mp_C$) over the entire simulation time) (Fig.. 5). Hence the reaction potential at each cell is Eq. 2:

$RP_C = \Sigma Mp_C$                    [2]

The reaction potential in the whole model is Eq. 3:

$RP_M = \Sigma RP_C$                    [3]

The indices C and M refer to cell and model, respectively.

To compare the variation in salinity ($\gamma$) of the different model cases the SD of each cell was summed up for each model case

($SumSD_{MC}$) divided by the number of cells (N) and normalized to the base case ($SumSD_{BC}$) Eq. 4:

$\gamma = [(Sum(SD_{MC})/N] / [(SumSD_{BC})/N]$                    [4]

$\gamma$ and $RP_M$ (normalized to the base case) serve as a measure for the dynamic character of the system and were used in a k-means cluster analyses to group the 24 cases (Fig.. 6).

**3 Results**

**3.1 Base and stable cases**

The simulations were transient and hence the flow field and the salinity distribution varied over time in all models except the steady state of the stable case (case 1). This temporal development is exemplarily shown for the base case in Fig. 2.





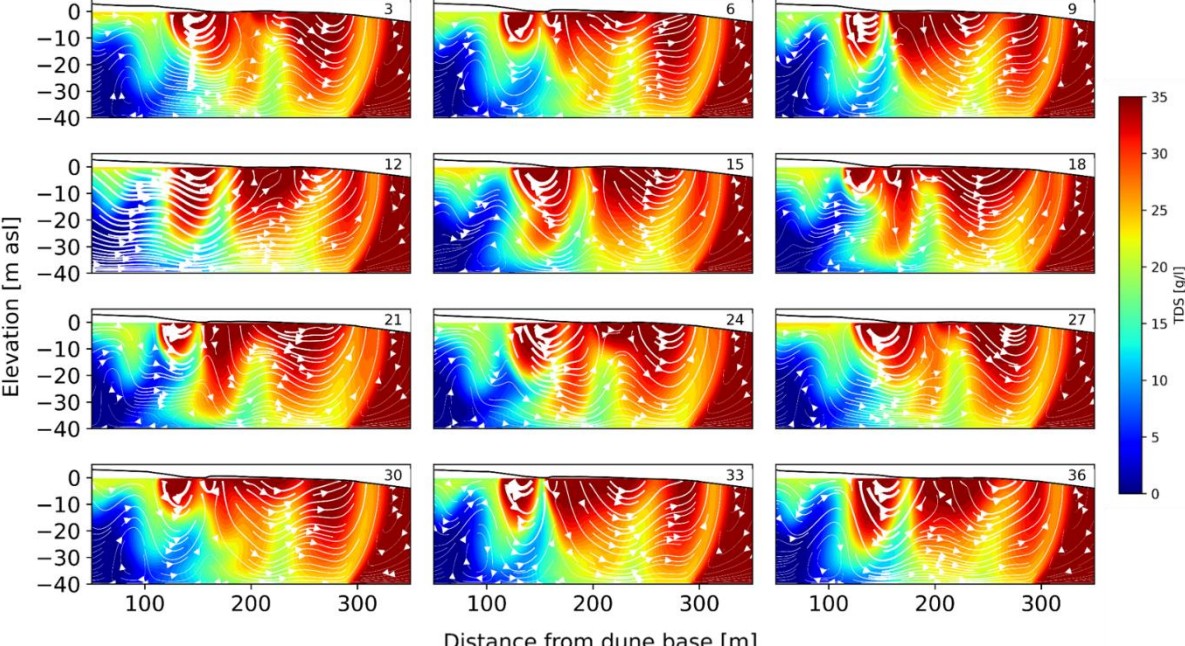

**Figure 2: Cross-sections of TDS distribution (colored areas) and flow field (white flow lines) for the base case, exemplarily shown for every three month of the last 3 years of the simulation period. The number in the corner refers to the month.**

**Base case**

The flow and salt dynamics in the base case (case 3) with a temporally varying topography and storm floods showed the formation of salt fingers and several USPs in front of the main USP moving through the subsurface. Consequently, salinities varied over time throughout the entire STE. Likewise, the flow velocities, flow directions and discharge locations varied over time. Flow was enhanced and flow lines were more horizontal in times of storm floods (e.g. Fig. 2, month 12). Flow lines were more undulating in regions with stagnant water outside the storm season. The SD of TDS was highest below the area of the MHWL that moved along the transect due to dynamic topography (Fig. 4, case 3). The Mp concentration was high down to 10 m asl below the storm flood affected area and deeper below the intertidal zone where the FDT and constantly moving USPs were mixing (Fig. 5, case 3). Compared to Greskowiak and Massmann (2021) the first layer was finer discretized to better accommodate for reactive transport but results otherwise showed in general the same flow and transport patterns.

**Stable topography cases**

A stable topography without storm floods (Fig. 3, case 1) led to the classical picture of a tidal STE (Robinson et al., 2006) with an USP, reaching down to 30 m, on top of a FDT that was pushed down by the infiltrating saltwater of the USP, discharging

near the MLWL while being pushed upward by the circulating SW. As would be expected in the stable case, the SD (Fig. 4, case 1) was very low, close to 0, and mixing controlled reactions took place along the fringes of the USP and the SW only (Fig. 5, case 1). Taking into account storm floods while maintaining a constant topography (case 2) led to changes of the flow regime and deformation of the USP (Fig. 3, case 2). At the same time, the reaction domain expanded and the mixing controlled reaction potential increased below the storm flood affected area (Fig. 5, case 2), where SD was higher (Fig. 4, case 2).

**3.2 Impact of aquifer parameters**

**Hydraulic conductivity**

A higher K (case 4) in this flux driven system resulted in high infiltration rates of saltwater in the USP and freshwater from inland. The freshwater was forced to move upwards, partly recirculating in front of a large, relatively stable (low SD) saltwater body (Fig. 3 and 4, case 4). A FDT did not form and freshwater discharged near the MHWL. The recirculating brackish water

in front of the saltwater body mixed constantly with inland freshwater and saltwater from the storm floods, resulting in an extensive zone with high reaction potential under the upper beach (Fig. 5, case 4). In contrast, a lower K (case 5) resulted in a widening of the FDT (Fig. 3, case 5). Saltwater was hindered to infiltrate and both the SD and the RP were high close to the SW and in the upper part of the aquifer influenced by storm floods (Fig. 4 and 5, case 5). A higher K anisotropy (case 6) led to the establishment of a relatively stable, wide FDT with uniform flow, with overall low SD (Fig. 3, 4, case 6). The high RP

zone was focused to the storm flood affected area the deeper USP and the wedge interface (Fig. 5, case 6). On the other hand, a lower K anisotropy (case 7) distorted the flow regime and led to strong fingering flow (Fig. 3, case 7). The zone of SD was small and focused below the MHWL (Fig. 4, case 7), where the RP was also high (Fig. 5, case 7).





**Figure 3:** Snapshots of the flow regime (white flow lines) and salinity distribution (coloured areas) for the 24 model cases at the end of the 20 years simulation time. The base case is framed in black. Thicker flow lines indicate higher flow velocities.





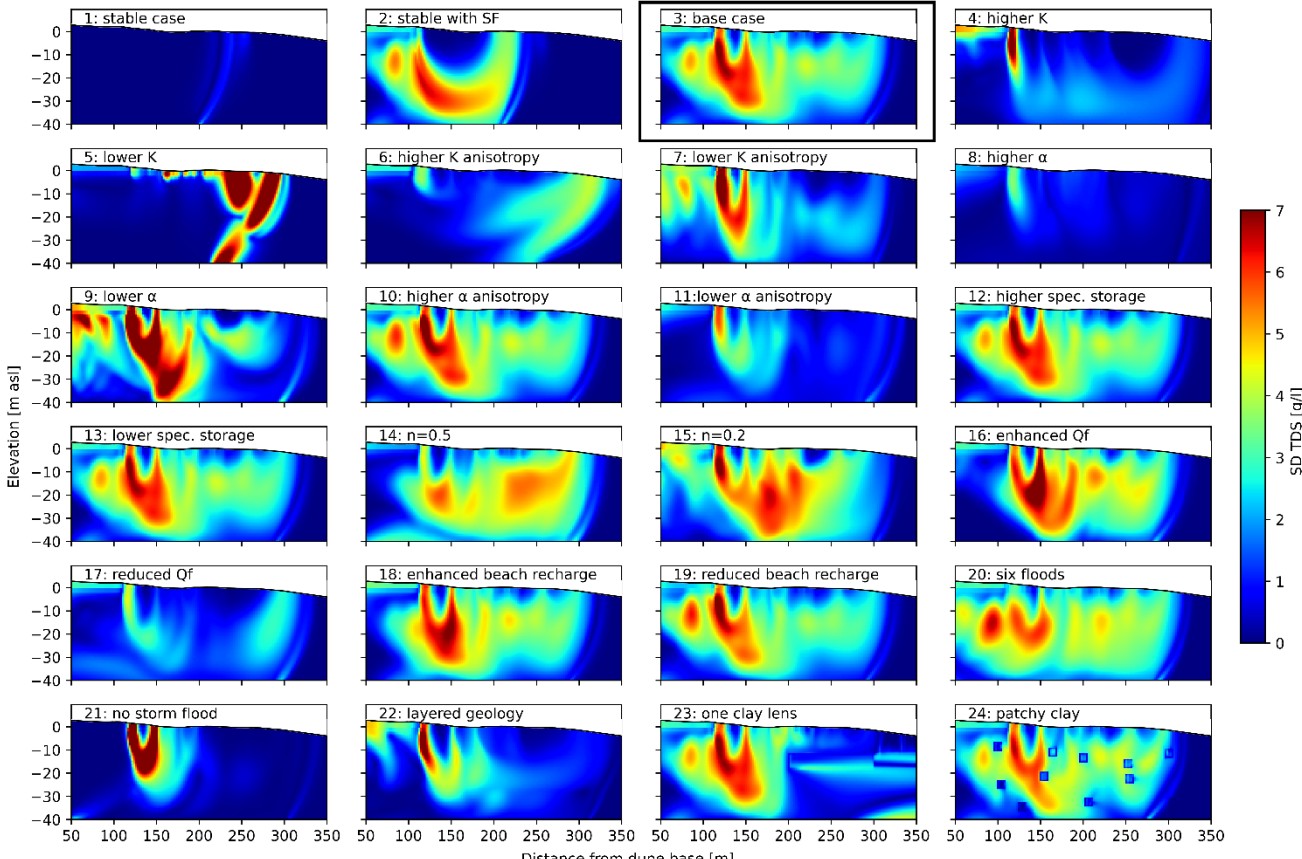

**Figure 4: Standard deviation (last 10 years of simulation time) of TDS over the last 10 years of simulation time for the 24 model cases. The base case is framed in black.**



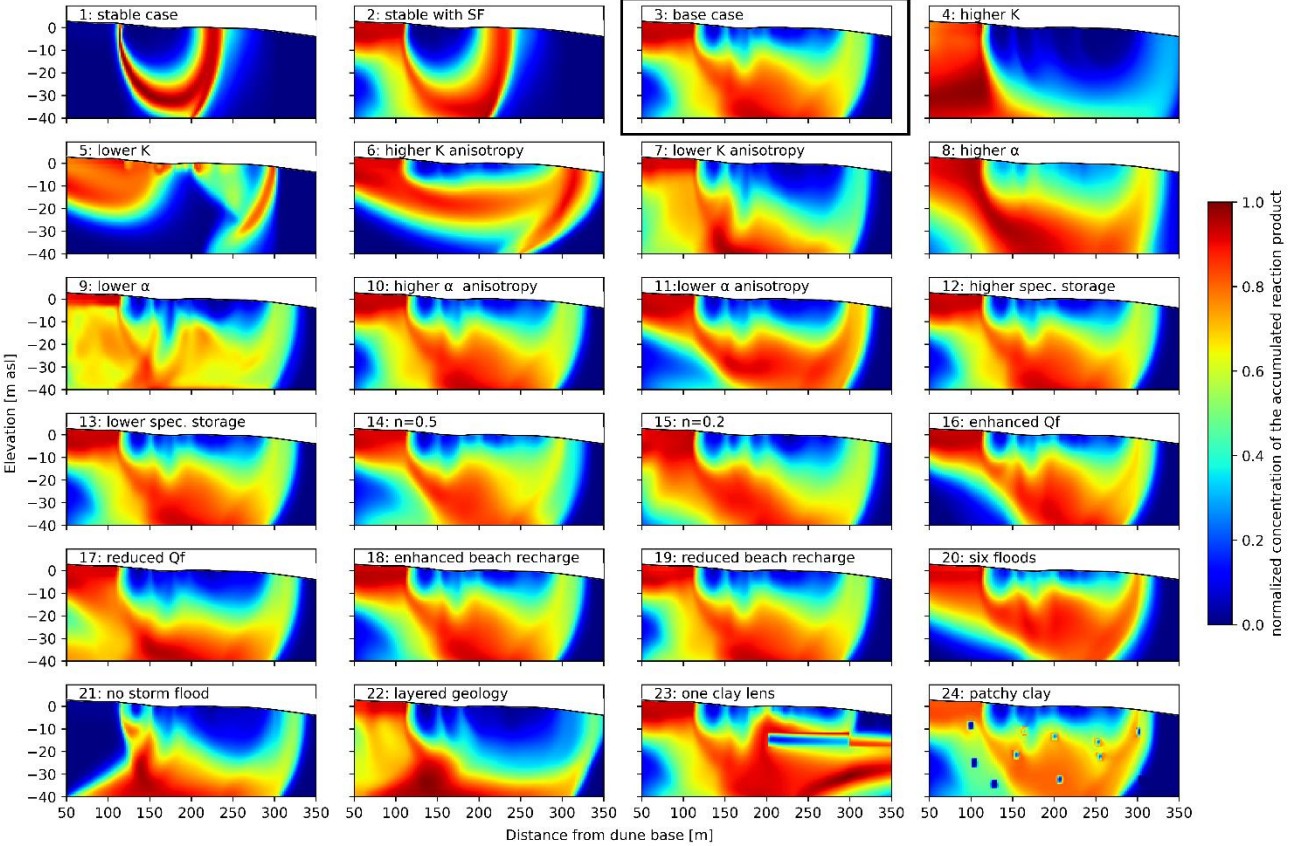

**Figure 5: Normalized concentration of the accumulated reaction product (Mp) indicating the reaction potential for all 24 model cases after 20 years of simulation time. The base case is framed in black.**

**Dispersivity**

The higher $\alpha_L$ case (Fig. 3, case 8) resembled the higher K case (Fig3, case 4). The higher dispersivity led to more mixing (wider zone of RP) and one big relatively stable dispersed USP formed (Fig. 4, 5, case 8). The FDT became more brackish and less distinct. The SD was generally low and the RP was large in the area bordering the USP and also much larger than in the high K case (case 4). A lower $\alpha_L$ (case 9) led to sharper contrasts between saline and freshwater bodies with wider mixing zones but with lower Mp concentrations (Fig. 3, 4 and 5, case 9).

A higher $\alpha$ anisotropy ($\alpha_T/\alpha_L$) (case 10) had only minor impact on the flow and transport patterns compared to the base case (Fig. 3, 4, case 10), except for a slightly more dispersed RP zone (Fig. 5, case 10). A lower $\alpha$ anisotropy ($\alpha_T/\alpha_L$) (case 11)



produced a more uniform flow regime below the upper beach with relatively stable TDS patterns indicated by small and focused SD (Fig. 3, 4, case 11) and a smaller, more focused RP zone along the fringe of the USP (Fig. 5, case 11).

**Specific storage**

The variation in specific storage (cases 12, 13) basically had no effect compared to the base case (Fig. 3, 4 and 5, cases 12 and 13).

**Porosity**

A higher porosity (case 14) increased the effective flow through area and reduced flow velocities (Fig. 3, case 14). Consequently, residence times were prolonged and hence the whole system was less mixed indicated by smaller SDs (Fig. 4,

case 14). RP was only slightly reduced compared to the base case (Fig. 5, case 14) because a reduction of mixing reduced the formation of MP but longer residences times enhanced the formation of MP. A reduction in porosity (case 15) had the opposite effect, i.e., less spore space caused faster flow-through, decreased residence times and enhanced mixing indicated by higher SDs and only slightly enhanced RP (Fig. 3, 4, 5, case 14).

**3.3 Impact of boundary conditions**

**Freshwater inflow**

Enhanced inflow from the inland freshwater boundary (case 16) had a freshening effect and resulted in a more uniform flow regime below the upper beach with higher TDS variation and a narrower RP zone, while the SD was more pronounced around the MHWL (Fig. 3, 4 and 5, case 16). A reduction in freshwater inflow (case 17) made the system more brackish (Fig. 3, case 17), resulted in less SD (Fig4, case 17) and caused a wider RP zone (Fig. 5 case 17).

**Beach recharge**

A variation in fresh beach recharge (cases 18, 19) showed effects mostly landwards of the USP below the storm flood affected upper beach. A higher beach recharge (case 18) led to stronger dilution of saltwater that previously infiltrated during storm flood events due to the higher groundwater recharge (Fig. 3, case 18). The SD was smaller than in the base case below the upper beach caused by the enhanced mixing, resulting in constantly brackish water (Fig. 4, case 18). Reduced beach recharge

(case 19) in turn showed sharper (higher concentration gradients) saltwater fronts from storm flood events which did, however, reached less deep (Fig. 3, case 19). The SD was higher in the shallow part of the upper beach caused by the downward mowing



sharper saltwater fronts (Fig. 4, case 19). While the overall RP was little affected by the variation in beach recharge compared to the base case, small differences were visible below the upper beach: more focused RP for enhanced recharge and less pronounced RP for reduced recharge, yet, impacting a larger area (Fig. 5, cases 18, 19).

**Storm floods**

Doubling the amount and frequency of the storm floods (case 20) in comparison to the base case enhanced the saltwater flux through the upper beach (Fig. 3, case 20). TDS increased as did the SDs (Fig. 3, 4, case 20) and the zone of higher RP reached further down below the upper beach (Fig. 5 case 20). Without storm floods (case 21) the system dynamics were driven only by the continuously changing topography, resulting in a movement of the intertidal zone and the USP across the shore (Fig. 3, case 21). The SDs were highest in the USP around the MHWL (Fig. 3, 4, case 21). The zone of high RP was restricted to the lower part of the aquifer, rising up below the MHWL (Fig. 5, case 21).

**Geological structure**

A layered geology (case 22) (alternating permeable sand layers) led to more mixing and dilution below the upper beach (Fig. 3, 4, case 22). The SDs were comparatively small and restricted to the MHWL. The high-RP zone was wider below the upper beach and covered here the entire aquifer thickness while it was restricted to the lower part of the aquifer below the intertidal zone.

A single, extensive clay lens (case 23) made a significant difference to the entire flow regime. On the landward side, the first 200 m of the subsurface looked similar to the base case (Fig. 3, case 23). In vicinity of the clay layer, several smaller USPs were restricted to the area above the clay layer, while flow was rather uniform below the clay layer. The SW was shifted further offshore (not visible in Fig. 3). The salt fingers moving towards the clay layer from onshore were distorted by the clay layer. The SDs were high below the upper part of the intertidal zone (Fig4, case 23) and the zone with high RP extended offshore (Fig. 5, case 23).

Patchy small clay lenses (case 24) distributed throughout the STE led to only small and very local deviations of groundwater flow and salinity patterns as well as local changes on SDs and RPs around the clay patches (Fig. 3, 4 and 5, case 24) while the overall picture was very similar to the base case.





### 3.4 Cluster analysis

A k-means analysis based on $\gamma$ and $RP_M$ of each model case normalized by to the same statistics of the base case clustered the model cases into three main groups (Fig. 6, Table 1). Cluster A (red circles) had a $\gamma$ (+/- 20%) and $RP_M$ (+/- 20%) similar to the base case (black cross). Cluster B included the less dynamic and more stable cases with a lower $\gamma$, reduced by 40-90%, and

lower $RP_M$, reduced by 30-60%, compared to the base case. Cluster B contained the cases with either changing topography only (and no storm floods, case 21) or only storm floods (and no changing topography, case 2) or neither (case 1) and the low K case (case 5). Cluster C was characterized by a lower $\gamma$ reduced by 40-60%, while keeping a $RP_M$ (+/- 20%) similar to the base case. Two outliers showed a significantly higher $\gamma$ and higher $RP_M$ (case 23: one clay layer) and lower $\gamma$ with higher $RP_M$ (case 8: higher $\alpha$) compared to the base case.


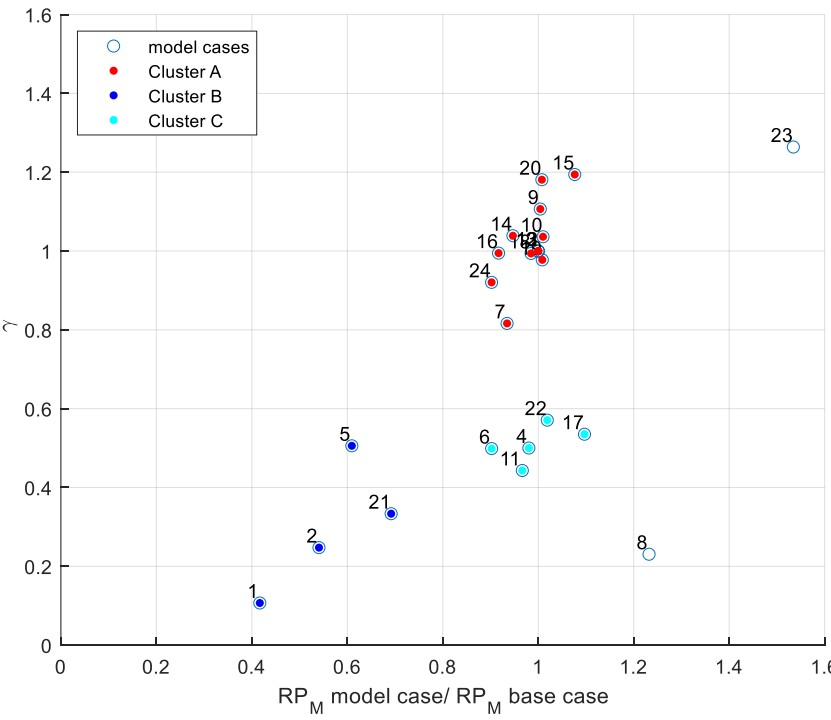

**Figure 6: Cluster analysis (k-means,3) of all model cases based on the models' variation in salinity ($\gamma$) and the sum of reaction product of each model ($RP_M$). Individual model values were normalized to the base case (located 1,1 in the diagram).**

**Hydrology and Earth System Sciences**
EGU

Discussions

## 4. Discussion


The purpose of this study was to identify the main drivers of the flow and transport dynamics in the subsurface of a high-energy beach by systematically changing aquifer parameters and boundary conditions, thereby evaluating their effect on the flow and transport regime as well as mixing controlled reaction potential. Conditions and parameters chosen are realistic for the barrier island Spiekeroog which is representative for a high-energy STE. We found that results are most sensitive to a

variable beach morphology, storm floods, hydraulic conductivities and dispersivity.

### 4.1 Flow and transport regime

A temporally changing morphology lead to the constant migration of the MHWL and MLWL on the beach transect. The intertidal zone width varies as does the USP and the discharge zones. This behaviour has previously been observed in field

studies targeting the shallow subsurface on Spiekeroog (Grünenbaum et al., 2020a; Waska et al., 2019). Findings of the present study indicate that constantly changing infiltration and exfiltration locations onset instable conditions that also drive the dynamics in the deep subsurface. Additionally, the beach surface, which is in situ sometimes straight and at other times forming a berm-trough-system (Grünenbaum et al., 2020a) due to sediment erosion-accretion, affects groundwater flow and salt transport as well as the mixing controlled reactions. This dynamic upper boundary results in flow paths and flow velocities

that vary continuously and, thereby enhance mixing in the subsurface. The seasonal storm flood events add saltwater to the upper beach that, together with the changing morphology, act as an effective pump pushing the saltwater, FDT and several USPs through the intertidal zone.

### 4.2 Mixing-controlled reaction potential

The dynamic groundwater flow and transport in the STE trigger mixing-controlled reactions.Anwar et al. (2014) found that oceanic forcing increases the freshwater-seawater mixing, enhancing nutrient transformation. Heiss et al. (2017) concluded that mixing-dependent processes are positively correlated with related to the size of the USP that in turn is affected by oceanic forcing and hydrogeological parameters. Our results show that in particular the dynamic beach morphology and the storm

floods drive subsurface flow and transport dynamics and result in changes of the location, shape, extent and intensity of what

we refer to as the mixing controlled reaction potential. Similarly, Greskowiak et al. (2023) concluded that a dynamic beach

morphology paired with storm floods were the main factors effecting redox-zones in the deep STE, which, however, are largely

a function of travel time rather than mixing.

### 4.3 Synthesis of the 24 model cases

We found that besides morphodynamics and storm floods also the horizontal and vertical hydraulic conductivity as well as

longitudinal and transversal dispersivity impact significantly on the shape, location, stability and extent of the USP, FDT and

mixing controlled reactions. However, our results also suggest that some factors only control the flow and transport regime

but not the reaction potential in the same way. For example, a higher longitudinal dispersivity (case 8) and a reduction in

freshwater inflow (case 17) caused a significantly reduced SD (Fig. 4) while the RP shows an overall similar shape compared

to the base case (Fig. 5).

Three main groups were identified in the k-means cluster analysis that best characterize the overall system behaviour as either

similar to the base case (cluster A), resulting in less variable salt patterns and less reactive conditions than the base case (cluster

B) or less variable salt patterns but still reactive conditions compared to the base case (cluster C). Changing the values of

hydraulic conductivity or dispersivity and their respective anisotropies made that simulations ended up in different clusters.

Hence subsurface dynamics are particularly sensitive to these parameters. Also, geological structures with a high K contrast

may strongly change flow, transport and mixing in the STE if they are large enough, as exemplarily shown for case 23 with

one large clay lens in the intertidal zone. Other parameters and boundaries or small-scale geological heterogeneities hardly

affected the system, but are still important to take into account when building site-specific models of STEs evaluated based on

point observations.

### 4.4 Study limitations

Our generic modelling approach has some limitations. We used a 2-D set up as did former studies at Spiekeroog (Beck et al.,

2017; Greskowiak and Massmann, 2021; Grünenbaum et al., 2020b; Röper et al., 2012) as well as at other sites (Anwar et al.,



2014; Heiss et al., 2017; Michael et al., 2016; Robinson et al., 2006). However, as our results show that morphological changes have a major impact on the salt distribution and considering that the beach morphology and associated head gradients are

variable along the shore (Geng and Michael, 2021; Knight et al., 2021; Paldor et al., 2022; Reckhardt et al., 2024), the subsurface flow in the STE is likely characterized by a 3-D flow component. Moreover, we used a measured, but linearly interpolated variable beach topography. In reality, rapid changes in morphology are likely to occur caused by short-term events (Karunarathna et al., 2018) such as storm floods or strong winds and consequently enhance the dynamics in the subsurface. Further, repeatedly changing the morphology over an annual cycle leads to cyclic groundwater flow and salt transport patterns

but in a real-world system the changes will be less regular.

In our generic modelling approach, we systematically changed single aquifer parameters and boundary conditions though some of these parameters may be correlated. At the same time, boundary conditions can be inter-dependent or affected by seasons. We chose this approach to better disentangle single effects and refrain from analysing combined cases.

To optimize the numerical effort with a daily time stepping we neglected waves and used the tide average approach as did e.g.

Vandenbohede and Lebbe (2007). However, previous studies showed that waves and tide-resolved approaches intensified the fresh-saltwater mixing, resulting in enhanced nutrient transformation and discharge of nutrients from the land to the sea (Anwar et al., 2014; Heiss et al., 2017). The use of a tide-average approach is also the reason why the change in storage parameters does not show any impact on groundwater flow and transport that would be expected in a real-world system. As demonstrated by Grünenbaum et al. (2020b), the effect is rather small on salinity and travel time distributions as tides produce only very

small back- and forth displacements of groundwater parcels within a tidal cycle, but it broadens the transition zone. We systematically started with a homogenous geology, showed a layered case and one with a thick clay lens and smaller clay patches instead of using a statistically generated heterogeneities as done by Michael et al. (2016). The heterogeneous distribution of other aquifer parameters, e.g. porosity (Meyer et al., 2018a), might also be of relevance, but was neglected here.

## 5. Conclusion

This study aimed at advancing the understanding of flow and transport processes in STEs under high-energy conditions. We systematically investigated the interplay of morphological changes and hydrodynamic boundary conditions paired with aquifer



properties of a high energy beach in a 2-D generic modelling approach with 24 model cases. Except for the stable case (case 1) which disregards morphodynamics and storm-floods, groundwater flow and transport was highly dynamic in all cases. The main factors controlling the subsurface dynamics were the changing beach morphology, storm floods, hydraulic conductivity

and dispersivity as well as major geological structures. The results highlight the need to account for the changing topography and storm floods when studying/modelling high energy beaches. Moreover, a good knowledge of the geological structures and parameter estimates is necessary. The flow and transport regimes are coupled to mixing-controlled reactions and hence transformation of nutrients in the subsurface. If disregarded, resulting net fluxes to and from the sea might be over- or underestimated.

In future, extending the model domain to 3-D while also accounting for morphodynamics could help to assess to what extent subsurface flow is 3-dimentional under high-energy conditions. The implementation of real data as, for example, currently collected from a beach observatory on Spiekeroog (Massmann et al., 2023), in model calibration will further help to better constrain and understand the biogeochemical functioning of high energy STEs. Here, we studied the extent of mixing controlled reactions in the subsurface by a simplified hypothetical reaction model. To move on to further study and quantify

in-situ biogeochemistry in the STE reactive transport modelling approaches targeting real-world reactions would be a valuable step forward.



**Acknowledgement**

This research study was conducted in the project Morphodynamics, sub-surface flow and transport (MA 3274/15–1) and

Reactive Transport (GR4514/3-1) within the research unit FOR 5094: The Dynamic Deep subsurface of high-energy beaches

(DynaDeep), funded by the German Research Foundation (Deutsche Forschungsgemeinschaft, DFG). We thank the entire

DynaDeep team as well as Patrick Hähnel and Lena Thissen for constructive discussions.

**Author contribution (CREdit)**


Rena Meyer: Writing – original draft, Visualization, Methodology, Investigation, Software, Formal analysis,
Conceptualization

Janek Greskowiak: Writing – review & editing, Visualization, Methodology, Investigation, Software

Stephan L. Seibert: Writing – review & editing, Visualization

Vincent Post: Writing – review & editing

Gudrun Massmann: Writing – review & editing, Conceptualization, Funding acquisition.

**Competing interests**

The authors declare no competing interests.

**Data & computer code availability**

The data used for the intertidal boundary condition as well as the FloPy python code for the base case are available in the

supporting material of Greskowiak & Massmann (2021).




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
