# Peer review of "Effects of boundary conditions and aquifer parameters on salinity distribution and mixing-controlled reactions in high-energy beach aquifers"

_Hydrology and Earth System Sciences, 2024_

## Author Response (AR1)

Manuscript **hess-2024-196:** "*Effects of boundary conditions and aquifer parameters on salinity distribution and mixing controlled reactions in high-energy beach aquifers.*"

Correspondence to Rena Meyer (rena.meyer@uni-oldenburg.de)

Author responses to Reviewer #1.

This manuscript aims to study the effects of beach morphological changes, hydrodynamic boundary conditions, and hydrogeologic properties on flow regimes, salt distribution, and the potential for mixing controlled chemical reactions in high-energy beach aquifers. The authors achieve this objective by building 2-D transient density-driven groundwater flow and transport numerical models that systematically explore boundary conditions and hydrogeological parameters in a changing beach morphology setting. The authors then conclude that changing beach morphology causes the migration of infiltration and exfiltration locations along the beach transect, leading to moving flow and salt patterns in the subsurface and enhancing mixing-controlled reactions.

I have read the manuscript with great interest. My overall opinion is that the manuscript is well-written but needs some moderate revisions and clarifications. Below, I have listed comments, hoping they may help improve the manuscript's quality.

Dear Reviewer #1 we appreciate your time and thank you for thoroughly reading and reviewing our manuscript. We appreciate the overall positive evaluation. Your comments are very valuable and improve the quality of our manuscript. We have responded to your comments point by point as indicated by Author Comments (AC) in blue, changes to the manuscript are indicated by "speech mark" and line numbers correspond to the clean version of the revised manuscript.

Kind regards, Rena Meyer

**Specific Comments**

1. Some clarification is needed regarding the 2-D numerical model approach.
    1. The beach boundary, where freshwater inflow occurs, is close to where flow, salinity, and mixing variations occur. Have the authors considered moving this boundary further inland to remove possible boundary effects?

    AC: We agree with the reviewer that the mixing effect would probably be enhanced if the boundary of fresh water inflow would lie further inland. We added this point to the limitation section of the study.

    Lines 384-392: "The location of the beach boundary to which freshwater inflow is assigned is in close proximity to where flow, transport and mixing variations occur. Consequently, it is likely to exert a boundary effect where increased freshwater inflow reduces TDS concentrations. Conversely, if the boundary were to lie further inland, TDS concentrations below the upper beach would be enhanced, as water is allowed to flow further inland during possible flow reversals. However, this only happens during extreme storm flood events, and as these are infrequent and short lived (1-2 days), there is not enough time for additional salt to accumulate as a result of the flow reversal. Furthermore, as this only occurs along the southern vertical boundary of the

models, we expect it to have an insignificant effect on the overall flow, transport and mixing patterns in the intertidal zone."

2. The description of the freshwater inflow boundary is missing from the methods section. Also, why is the value of 0.5 m$^3$/d/m decided for most cases? Is this value related to the groundwater discharge in the area where the model is based?

AC: The value for the fresh groundwater inflow of 0.5 m$^3$/d/m was be added to the methodology section. It relates to the subsurface freshwater inflow from the islands' freshwater lens. The value was estimated by Beck et al. (2017) and relates to the fresh groundwater entering from the South with a flow rate of 0.5m$^3$ per day and meter of shoreline and was estimated from the approximate distance to the groundwater divide and the approximate recharge rate.

Lines 127-129 : " Freshwater (salt concentration = 0 g/l) entering from the interior of the island (freshwater lens) along the southern vertical boundary was prescribed using a specified flux of 0.5 m3/per day per meter coastline as estimated by Beck et al. (2017), and distributed uniformly across the cells of the first column (Fig. 1)."

3. The authors mention that meteoric groundwater recharge was applied at the upper beach slope above the mean high water line (MHWL) (lines 123 to 124). What is the value of this recharge (350 or 400 mm/a)? These values are missing in Table 1.

AC: The meteoric groundwater recharge at the upper beach is 400 mm/a. We added the value to the methodology section.

Line 130: "Meteoric groundwater recharge of 400 mm/a was applied at the upper beach above the MHWL"

4. What is the initial concentration distribution in the model? Did the authors run a warm-up period before calculating the variations?

AC: We understand the concern of the reviewer about the effect of the initial distribution of TDS. The initial TDS concentrations were 0 g/l landwards from the mean low water line and 35 g/l towards the seaside. This is explained in the methodology section.

The total simulation time was 20a. The calculation of the variability of TDS (SD of TDS) was based on the last 10a of simulation time, when the initial concentrations have no significant influence on the dynamic flow and transport patterns any more. Thus, the first 10a simulation time serve as model spin-up period to avoid any impact of the initial conditions on the final model results. We extented the explanation in the methodology sections accordingly.

 Lines 186 – 188: "The decision to assess SD TDS and RP$_c$ based on the final 10 years of the simulation period was taken in order to circumvent the potential influence of the initial distribution of TDS, R$_s$ and R$_f$. Hence, the first 10 years of the simulation period serve as model spin-up."

5. In my mind, when the beach morphology changes, part of the model geometry also changes. How do the authors estimate the variations in salinity and mixing patterns with varying geometry?

AC: The topographical changes were not applied to the model grid, this remained constant over time. The topographical changes were used to calculate the hydraulic heads that are applied to the sea boundary above the LWL (Fig.1). This approach was also used by Greskowiak and Massmann (2021).
We extented the explanation in the methodology section accordingly, see AC to the next point.

1. Throughout the manuscript, the authors highlight the importance of changes in beach morphology in the dynamics of upper salinity plumes (USPs). However, the modeling choice to account for these changes needs further clarification.

   1. It is unclear how the surface interpolation was made, and given that the authors express the importance of this feature in the modeling choice, more information should be provided on this matter. Thus, I recommend adding more information on the methods of how the interpolation was made and refining the plot in Figure 1 so that the reader can picture how beach morphology changes over time.

AC: We understand the necessity for clarification of the interpolation method of the beach topography.
The interpolation of the beach surface was performed as proposed by Greskowiak and Massmann (2021) and their supporting information. The five cross-shore LIDAR scan profiles (as shown in Fig. 1b, obtained for Feb-July), were interpolated to daily increments over a six-month period. Subsequently, the topography was varied in daily steps over half a year and then reversed for the second half of the year. The resulting annual topography was then employed recursively over the 20a simulation period. The resulting time series of daily topography was then used to calculate the hydraulic heads that were applied to the GHB in the intertidal zone.

We extended the explanation of the interpolation procedure in the manuscript by:

Lines 134-140: "In the intertidal zone (between the MLWL and the upper beach affected by storm floods) the beach surface was interpolated according to the methodology described by Greskowiak and Massmann (2021). Five cross-shore LIDAR scan profiles (as shown in Fig. 1b, obtained for Feb-July), were sampled at 1m resolution and interpolated to daily increments over a six-month period. The topography was then varied in daily increments for six month, and reversed for the second half of the year. The resulting annual topography was applied recursively over the 20a simulation period. Daily topography time series were then used to calculate the hydraulic heads using the tide-average head approach, which were assigned to the sea boundary in the intertidal zone (Fig. 1a, grey box)."

As suggested by the reviewer, Figure 1 was updated to better demonstrate how the intertidal topography changes over time:

New Figure 1

[Figure]

Figure 1: (a) Model setup with dimensions, boundary conditions and parameters, (colors indicate TDS distribution: red = saline, blue = fresh). The thin grey box encompasses the part of the sea boundary where changes in beach morphology occur. (b) Four LIDAR scans (Grünenbaum et al., 2020a) of intertidal topography (colored lines) and respective interpolated topographies (grey lines in 10-days increments). (c) – (f) Four different geological settings, note K = 0.005 m/d for the clay lens and clay patches (dark blue).

2. This reviewer understands the study's limitations. Between lines 351 and 355, the authors reference the limitations of linearly interpolating the beach morphology. However, how do the authors reconcile the idea of cycling beach morphology when changes in the hydrologic forcings and hydraulic properties occur? Have the authors considered changing the order of the interpolation in the beach profiles to see if different patterns emerge? Also, the authors only considered a stable case with and without storm floods. Have they considered exploring the variations in hydrogeologic parameters and boundary conditions with stable cases to compare them with the changes in beach morphology?

AC: As we outline in the limitation section we recognized the issue about temporally linear interpolation and cycled morphology which most probably underestimates the mixing in the STE sections. The suggestion of the reviewer to change the order of the interpolation is a good idea and we will consider it for future studies. Given that we already have an extended set of many model cases, we refrain from further extending our model suite in this study. However, we brought up the reviewers point in the outlook that it would be a valid step for future studies.

Lines 417-420: "The implementation of real data, including a high-resolution beach topography, as, for example, currently collected from a beach observatory on Spiekeroog (Massmann et al., 2023), in model calibration will further help to better constrain and understand the biogeochemical functioning of high energy STEs."

The variations in hydrogeological parameters and boundary conditions with stable morphologies have been studied before, e.g. Michael et al., (2016). Therefore, we decided to use the model of Greskowiak and Massmann (2021) as base case and studied the effect of single changes to boundary conditions and parameters to evaluate their individual effect on the flow, transport and mixing dynamics and refrain from considering combinations of the parameters and boundary conditions. This aspect was mentioned in the limitation section:

Lines 392-395: "In our generic modelling approach, we systematically changed single aquifer parameters and boundary conditions though some of these parameters may be correlated. At the same time, boundary conditions can be inter-dependent or affected by seasons. We chose this approach to better disentangle single effects and refrain from analyzing combined cases."

2. Regarding mixing-controlled reactions, the authors used the initial salinity patterns for the Rf and Rs concentration patterns. This modeling choice could create misleading mixing patterns because these chemical constituents were assumed conservative, and the model needs to stabilize first. So, did the authors consider warming up the model before setting the initial concentration patterns?

AC: The reviewer is right that the initial distribution of Rf and Rs could have had an effect on the final patterns of their mixing product (Mp). In order to resolve this issue, we have now subtracted the accumulated mixing concentration after 10a (when the initial concentrations of TDS did not influenced final salinity patterns any longer) from the final (20a) simulation concentration. The general mixing patterns look very similar to the previous results, as presented in the new Figure 5 (see below). However, slight changes in normalized concentrations are visible (Fig. 5). Therefore, we decided to follow the reviewers recommendations and updated Figure 5 and also re-calculated the cluster analysis (Fig. 6) which also showed slight differences. Moreover, we modified the corresponding text about the cluster analysis. However, the overall changes were only minor and the general picture and conclusion driven are still valid.

The methodology section was extended by:

Lines 186-188: "The decision to assess SD TDS and $RP_c$ based on the final 10a of the simulation period was taken in order to circumvent the potential influence of the initial distribution of TDS, $R_s$ and $R_f$."

Update to Figure 5:

[Figure]

**Figure 5: Normalized concentration of the accumulated reaction product (Mp) indicating the reaction potential for all 24 model cases over the last 10 years of simulation time. The base case is framed in black. The letter in the upper right corner refers to the cluster group (cf. Tab. 1, Fig. 6).**

The cluster analysis in the result section was modified to:

Lines 316-323: "A k-means analysis based on $\gamma$ and $RP_M$ of each model case normalized to the same statistics of the base case, resulted in the clustering of the model cases into three main groups (Fig. 6, Table 1). Cluster A (red circles) had a $\gamma$ (+/- 20%) and $RP_M$ (+/- 20%) similar to the base case (located at the coordinates 1,1 in the diagram in Fig. 6). Cluster B included the less dynamic and more stable cases with a lower $\gamma$, reduced by 40-98%, and lower $RP_M$, reduced by 30-70%, compared to the base case. Cluster B contained the cases with either changing topography only (and no storm floods, case 21) or only storm floods (and no changing topography, case 2) or neither (case 1) and the low K case (case 5). Cluster C was characterized by a lower $\gamma$ reduced by 40-80%, while keeping a $RP_M$ (+/- 20%) similar to the base case. One outlier, case 23 with one clay layer, showed a significantly higher $\gamma$ and higher $RP_M$ compared to the base case."

Update to Figure 6:

[Figure]

**Figure 6: Cluster analysis (k-means,3) of all model cases based on model variation in salinity (γ) and the sum of reaction product of each model (RP$_M$). Individual model values were normalized to the base case (located at 1,1 in the plot).**

**Technical Corrections**

Besides the comments described above, I have a few technical recommendations for the manuscript.

1. In line 17 (in the abstract), I recommend being more specific in the sentence: "The objective was to investigate their individual effects…," whose individual effects?

   AC: We modified the sentence in the abstract to be more specific to:

   Lines 17-19: "The objective was to investigate the individual effects of boundary conditions and hydrogeological parameters on flow regime, salt distribution, and potential for mixing controlled chemical reactions in a system with a temporally-variable beach morphology."

2. I recommend rewording the sentence between lines 55 and 58. It is difficult to understand what the other studies found and their limitations.

   AC: We modified the sentence for the sake of clarity to:

   Lines 56-59: "These studies provided field evidence supported by numerical models for the - at least temporal - occurrence of more than one USP for different beach slopes (Abarca et al., 2013) or typical sandy beach surfaces like runnel-ridge (Grünenbaum et al., 2020b) and through-berm (Robinson et al., 2006) structures."

3. A comma is missing between "…model Greskowiak…" in line 62.

   AC: The comma was added.

4.  Review the sentence between lines 67 and 69. It sounds redundant.

    AC: The sentence was rephrased to:

    Lines 67-70: "Greskowiak et al. (2023) concluded that redox zone dynamics in the STE are strongly affected by beach morphodynamics. While some redox reactions take place in the USPs and storm flood affected area, mixing-controlled reactions driven by mixing of two solutes in different end members occur in the fringes of the USP and at the SW interface (Heiss et al., 2017)."

5.  There is a double comma in line 86.

    AC: We removed it.

6.  The term **variability** of total dissolved solutes (TDS) (in line 88) is ambiguous. Are there variations in the concentration values of TDS, the spatial distribution or which variability?

    AC: We understand the reviewer that clarification is needed for the term variability of TDS and changed the wording to be more specific to:

    Lines 88-93 "[…] the aims are to systematically evaluate the effect of […] on (1) the flow regime, (2) the distribution of total dissolved solids (TDS) and their temporal concentration variability calculated as the standard deviation of TDS, [...]"

7.  The statement **generic modelling approach** (in line 93) is also ambiguous. What do the authors mean by "generic modelling"?

    AC: We understand that some clarification about the term generic modelling is needed. We consider our modelling approach "generic" in contrast to a field-site model, given that our objective is not to resemble the hydrogeological situation at a specific location in detail. It is not our intention in this study to utilize any field observations, for example for the purpose of calibrating the models. In contrast to a site-specific approach, our methodology entails a systematic alteration of boundary conditions and hydrogeological parameters, thereby facilitating a more comprehensive understanding of the physical processes occurring in STEs. However, we chose the model geometry and hydro-meteorological forcings loosely based on Spiekeroog's conditions because it is a well-studied site and representative for barriers islands under similar high energy settings (e.g., in terms of tidal amplitude and significant wave height), for example those occurring along the Wadden Sea (Netherlands, Germany and Denmark).

    We included  a short explanation of our understanding of a "generic modelling approach" to the last part of the introduction:

    Lines 83-88: "The objective of the present study is to investigate the interplay of morphological changes and hydrodynamic boundary conditions paired with aquifer properties in the subsurface of high energy beaches in a 2-D density-dependent generic modeling approach. Our model is considered 'generic' because it doesn't aim to replicate specific site conditions using field data or calibration. Instead, boundary conditions and parameters a varied to explore physical processes in STEs. While based

on Spiekeroog's conditions, the model aims to represent barrier islands in high-energy environments like the Wadden Sea, rather than a specific location."

8. Line 95 is missing a space in "…conditions(Hayes, 1979)…."

   AC: The space was added.

9. Consider rewording the sentence and citation between lines 100 and 101.

   AC: We rephrased the respective sentence to:

   Lines 105-106: "A slightly adapted version of the model by Greskowiak and Massmann (2021) serves as the base case for the 24 simulation cases in this study."

10. In Figure 1,
    1. There is a small box in (a) that is not described in the caption. Is this the area where changes in beach morphology occur? If so, I recommend stating it in the caption.

    AC: The reviewer is right that the box encompasses the area where the variable beach morphology was applied. We will extend the caption of Figure 1 respectively. We included an update to Figure 1 (see above)

    Caption of Figure 1: "Figure 1: (a) Model setup with dimensions, boundary conditions and parameters, (colors indicate TDS distribution: red = saline, blue = fresh). The thin grey box encompasses the part of the sea boundary where changes in beach morphology occur. (b) Four LIDAR scans (Grünenbaum et al., 2020a) of intertidal topography (colored lines) and respective interpolated topographies (grey lines show 10-day increments). (c) to (f) Four different geological settings, note $K_h = 0.005$ m/d for the clay lens and clay patches (dark blue)."

    2. Is the hydraulic conductivity presented in (c) through (f) the horizontal ($K_h$) or the vertical ($K_v$) component?

    AC: The horizontal hydraulic conductivity is presented in Fig.1 c-f . We added it to the figure's colorbar (see updated Figure 1 above).

    3. There is a space missing between "…scans(Grünenbaum et al., 2020a)…" in the caption.

    AC: We added the space.

11. The concentration of saltwater is missing in line 134.

    AC: The concentration of saltwater of 35g/l was added.

12. This might conflict with the journal's requirements, but I suggest changing the equations to a math format as it is easier to read.

    AC: We considered this in consultation with the Journal requirements.

13. In Table 1,
    1. What do the bold values mean? Are they to highlight the changing parameters? If so, I recommend stating this in the caption and keeping the bold values consistent throughout the rows.

    AC: The reviewer is right, these are the values that were changed compared to the base case. We included this explanation to the caption.

    Change to caption of Table 1: "Table 1: Aquifer properties and boundary conditions for the 24 model cases. Numbers in bold highlight the changes compared to the base case. Note that the base case is a case with a dynamic topography resembling the model by Greskowiak and Massmann (2021)."

    2. What is the `-"-` symbol in $\alpha_L$ in cases 4 and 5?

    AC: The symbol was wrong, and the value of the longitudinal dispersivity of 2m was added instead.

    3. What is the acronym SF? Is it Storm Flood? If so, please state it somewhere in the text.

    AC: The reviewer is right, SF is the acronym for storm flood. We added it to the text in line 74.

    4. I recommend capitalizing the "based case" in the description of case 3 to be consistent with the rest of the table.

    AC: We changed it as recommended by the reviewer.

    5. I recommend correcting the spacing in the description of case 15 to be consistent.

    AC: The spacing was changed as recommended.

14. I consider the clustering analysis a neat exercise for examining similar models. However, switching between Figure 6 and Figures 3 through 5 can be cumbersome. I recommend adding boxes around the subplots in Figures 3 to 5 that represent the colors of the clusters; that way, it is easier for the reader to visualize which models are clustered together.

    AC: We understand the need for a better recognition of clusters in the Figures 3, 4 and 5. Instead of adding colored boxes we will add the cluster groups name "A, B, C", in the upper right corner for each subfigure in Figures 3 to 5. These cluster groups are also referred to in Table 1. We extended the Figure captions respectively by

    Changes to the caption of Figures 3, 4, 5: "The letter in the upper right corner refers to the cluster group (cf. Tab. 1, Fig. 6)."

**References**

Abarca, E., Karam, H., Hemond, H.F., Harvey, C.F., 2013. Transient groundwater dynamics in a coastal aquifer: The effects of tides, the lunar cycle, and the beach profile. Water Resour. Res. 49, 2473–2488. https://doi.org/10.1002/wrcr.20075

Beck, M., Reckhardt, A., Amelsberg, J., Bartholomä, A., Brumsack, H.J., Cypionka, H., Dittmar, T., Engelen, B., Greskowiak, J., Hillebrand, H., Holtappels, M., Neuholz, R., Köster, J., Kuypers, M.M.M., Massmann, G., Meier, D., Niggemann, J., Paffrath, R., Pahnke, K., Rovo, S., Striebel, M., Vandieken, V., Wehrmann, A., Zielinski, O., 2017. The drivers of biogeochemistry in beach ecosystems: A cross-shore transect from the dunes to the low-water line. Mar. Chem. 190, 35–50. https://doi.org/10.1016/j.marchem.2017.01.001

Greskowiak, J., Massmann, G., 2021. The impact of morphodynamics and storm floods on pore water flow and transport in the subterranean estuary. Hydrol. Process. 35, 1–5. https://doi.org/10.1002/hyp.14050

Greskowiak, J., Seibert, S.L., Post, V.E.A., Massmann, G., 2023. Redox-zoning in high-energy subterranean estuaries as a function of storm floods , temperatures , seasonal groundwater recharge and morphodynamics. Estuar. Coast. Shelf Sci. 290, 108418. https://doi.org/10.1016/j.ecss.2023.108418

Grünenbaum, N., Ahrens, J., Beck, M., Gilfedder, B.S., Greskowiak, J., Kossack, M., Massmann, G., 2020a. A Multi-Method Approach for Quantification of In- and Exfiltration Rates from the Subterranean Estuary of a High Energy Beach. Front. Earth Sci. 8, 1–15. https://doi.org/10.3389/feart.2020.571310

Grünenbaum, N., Greskowiak, J., Sültenfuß, J., Massmann, G., 2020b. Groundwater flow and residence times below a meso-tidal high-energy beach: A model-based analyses of salinity patterns and 3H-3He groundwater ages. J. Hydrol. 587, 124948. https://doi.org/10.1016/j.jhydrol.2020.124948

Heiss, J.W., Post, V.E.A., Laattoe, T., Russoniello, C.J., Michael, H.A., 2017. Physical Controls on Biogeochemical Processes in Intertidal Zones of Beach Aquifers. Water Resour. Res. 53, 9225–9244. https://doi.org/10.1002/2017WR021110

Michael, H.A., Scott, K.C., Koneshloo, M., Yu, X., Khan, M.R., Li, K., 2016. Geologic influence on groundwater salinity drives large seawater circulation through the continental shelf. Geophys. Res. Lett. 43, 10,782-10,791. https://doi.org/10.1002/2016GL070863

Robinson, C., Gibbes, B., Li, L., 2006. Driving mechanisms for groundwater flow and salt transport in a subterranean estuary. Geophys. Res. Lett. 33, 3–6. https://doi.org/10.1029/2005GL025247

Manuscript **hess-2024-196**: "*Effects of boundary conditions and aquifer parameters on salinity distribution and mixing controlled reactions in high-energy beach aquifers.*"

Correspondence to Rena Meyer (rena.meyer@uni-oldenburg.de)

Author responses to Reviewer #2.

The main contribution of this work is that the authors showed that geomorphological changes drive changes in salinization and reaction potential in beach aquifers using general 2-D numerical models. They showed that there is greater mixing of fresh and saltwater due to geomorphic change and this leads to greater reaction potential. They also demonstrated that hydraulic conductivity, dispersivity, and storm floods control the extent of the mixing zone and salinization under changing geomorphic conditions.

The authors provide an important contribution by incorporating geomorphic change into coastal groundwater models. The paper is technically sound and well written, but it could benefit from a few clarifications in the methods section. Therefore, I recommend minor revisions in accordance with the comments below.

Dear Reviewer #2 we thank you for reviewing our manuscript and appreciate the overall positive assessment. We believe that your comments and suggestions help to significantly improve our paper. We have responded to your comments point by point as indicated by author comments (AC) in blue, changes to the manuscript are indicated by "speech mark" and line numbers correspond to the clean version of the revised manuscript.

Kind regards, Rena Meyer

Minor Comments

Line 96 What is the datum that MHWL and MLWL are referenced to?

AC:. Mean high water line (MHWL) and mean low water line (MLWL) relate to the 10-years mean from 2010 to 2020, measured at the tidal station Wangerooge North (https://www.pegelonline.wsv.de/gast/stammdaten? pegelnr=9420030) and referenced to the normal sea level (Normalhöhennull, NHN) reference system. We added this information to the methodology section.

Lines 100-101: "The mean high water line (MHWL), referring to the ten-year mean from 2010 to 2020 (Pegelonline, 2022), is located at 1.35masl and the mean low water line (MLWL) is at -1.35masl, referenced to normal sea level (NHN)."

Line 122 What is the specified flux that was used?

AC: We added the specific flux of 0.5 m3/d/m to the respective sentence.

Lines 127-129: "Freshwater (salt concentration = 0 g/l) entering from the islands' inland (freshwater lens) along the vertical Southern boundary was prescribed using a specified flux of 0.5 m3/day per meter shoreline as estimated by Beck et al., (2017) and was uniformly distributed across the cells of the first column (Fig. 1)."

Line 123 What is the flux for the meteoric recharge boundary?

AC: The meteoric groundwater recharge at the upper beach is 400 mm/a. We added the value to the methodology section.

Line 130 "Meteoric groundwater recharge of 400 mm/a was applied at the upper beach above the MHWL."

Lines 124 Why was a general head boundary used?

AC: The general head boundary was used to avoid numerical problems that may occur when a constant head boundary is used in a highly transient system where the assignment of boundary condition would change with each time step (due to the temporal changes in beach morphology). To mimic a constant head boundary we, however, chose a very high conductance of $1000 m^2$/day.

We included the reasoning to the methods section.

Line 130-134: "A general head boundary (GHB) with a high conductance of 1000 m²/d was specified along the seaside and intertidal zone (Fig. 1) to ensure good aquifer connection. The hydraulic head of the GHB boundary was set to 0 below the MLWL. This approach helped avoid numerical issues that could arise from using a constant head boundary in a highly transient system, where boundary conditions might change due to shifts in beach morphology."

Lines 128-130 How was the linear interpolation performed from the lidar scans (i.e. was each point in the domain sampled from the lidar scan or was it sampled at a different resolution), and how was the morphological change represented in the model (i.e. how was the interpolation done between each morphological realization, if erosion occurred was salt mass removed from the model and if deposition occurred was it assumed to be saline or fresh)?

AC: We understand the need to expand our explanation of how the variable beach morphology was derived from the LIDAR profiles. Each meter was sampled from the LIDAR scan. Interpolation between the morphological realizations was linear in daily steps, hence sudden erosion/accretion events are not displayed (therefore no additional salt mass was added or removed), as we outline in the limitations of our study.

We extended the description of the interpolation of the morphology in the methodology section:

Lines 134-140: "In the intertidal zone (between the MLWL and the upper beach affected by storm floods) the beach surface was interpolated according to the methodology described by Greskowiak and Massmann (2021). Five cross-shore LIDAR scan profiles (as shown in Fig. 1b, obtained for Feb-July), were sampled at 1m resolution and interpolated to daily increments over a six-month period. The topography was then varied in daily increments for

six month, and reversed for the second half of the year. The resulting annual topography was applied recursively over the 20a simulation period. Daily topography time series were then used to calculate the hydraulic heads using the tide-average head approach, which were assigned to the sea boundary in the intertidal zone (Fig. 1a, grey box)."

Line 133 Why was the simulated salinity assigned to the water discharging across the ocean boundary?

AC: With this we consider a so-called non-dispersive flux boundary condition. It's a 3$^{rd}$-type transport boundary condition, where solute mass is transported out of the model domain via the product of discharging water and its respective solute concentration.

Lines 145-148: "In the intertidal zone at the GHB, saltwater inflow and outflow were modelled using non-dispersive flux boundaries. A solute concentration of 35 g/L was assigned to the inflowing saline water and the simulated concentration to the outflowing water. This third type of transport boundary condition transports solute mass out of the model domain based on the product of the discharging water and its respective solute concentration."

Line 134 Was the simulation run to a steady-state salinity distribution before running the transient model?

AC: We did not run a steady-state salinity in our transient models. However, after 10 years of simulation the initial distribution of salinities (as TDS) and reactant were insignificant. Therefore, we chose to only use the last ten years of simulation period for our analysis to make sure that the initial conditions do not influence on our results and conclusion.

We extented the explanation in the methodology section accordingly:

Lines 186-188: "The decision to assess SD TDS and $RP_c$ based on the final 10 years of the simulation period was taken in order to circumvent the potential influence of the initial distribution of TDS, $R_s$ and $R_f$. Hence, the first 10a of the simulation period serve as model spin-up."

153-154 It seems like the initial distribution of the reactants is unrealistic given that there was no spin up to a steady-state salinity distribution. This should be mentioned as a limitation.

AC: We understand the necessity for clarification of the influence of the initial distribution of the reactants. Reviewer #1 made a similar comment. We repeat our AC to Reviewer #1 here for an easier reading:
The reviewer is right that the initial distribution of Rf and Rs could have had an effect on the final patterns of their mixing product (Mp). In order to resolve this issue, we have now subtracted the accumulated mixing concentration after 10a (when the initial concentrations of TDS did not influenced final salinity patterns any longer) from the final (20a) simulation concentration. The general mixing patterns look very similar to the previous results, as presented in the new Figure 5 (see below). However, slight changes in normalized concentrations are visible (Fig. 5). Therefore, we decided to follow the reviewers recommendations and update Figure 5 and also re-calculated the cluster analysis (Fig. 6) which also shows slight differences. Moreover we modified the corresponding text about the

cluster analysis. However, the overall changes were only minor and the general picture and conclusion driven are still valid.

The methodology section was extended by:

Lines 186-188: "The decision to assess SD TDS and $RP_c$ based on the final 10 years of the simulation period was taken in order to circumvent the potential influence of the initial distribution of TDS, $R_s$ and $R_f$. Hence, the first 10a of the simulation period serve as model spin-up."

Update to Figure 5:

[Figure]

**Figure 5: Normalized concentration of the accumulated reaction product (Mp) indicating the reaction potential for all 24 model cases over the last 10 years of simulation period. The base case is framed in black. The letter in the upper right corner refers to the cluster group (cf. Tab. 1, Fig. 6).**

Update to Figure 6:

[Figure]

The section about the cluster analysis in the result section was modified to:

Lines 316-323: "A k-means analysis based on γ and RP$_M$ of each model case normalized to the same statistics of the base case, resulted in the clustering of the model cases into three main groups (Fig. 6, Table 1). Cluster A (red circles) had a γ (+/- 20%) and RP$_M$ (+/- 20%) similar to the base case (located at the coordinates 1,1 in the diagram in Fig. 6). Cluster B included the less dynamic and more stable cases with a lower γ, reduced by 40-98%, and lower RP$_M$, reduced by 30-70%, compared to the base case. Cluster B contained the cases with either changing topography only (and no storm floods, case 21) or only storm floods (and no changing topography, case 2) or neither (case 1) and the low K case (case 5). Cluster C was characterized by a lower γ reduced by 40-80%, while keeping a RP$_M$ (+/- 20%) similar to the base case. One outlier, case 23 with one clay layer, showed a significantly higher γ and higher RP$_M$ compared to the base case."

Line 180 I think it would help the reader to describe the difference between the base case and the stable case before discussing the results.

AC: We added a few sentence to describe the difference between base case and stable case at the beginning of the results section.

Lines 201-204: "The model proposed by Greskowiak and Massmann (2021) which incorporates a transient beach morphology and three storm floods was employed as *base case* in the present study. Boundary conditions and aquifer parameters were then varied to identify their individual influence on flow and salt transport as well as mixing-controlled reaction potential. In the *stable case* the average beach topography from the base case and no storm floods were considered."

Line 193-195 I am confused by Figure 5 case 3. How is it normalized? If it is normalized to the base case then there should be no variation in the base case figure. A description of the normalization should be included in the methods.

AC: We understand the necessity for clarification of the normalization procedure that yields the concentration illustrated in Figure 5. The concentrations of the accumulated reactant are normalized to the maximum reactant concentration (M$_p$) across all model versions. We provided a more comprehensive description of the normalisation in the methodology section.

Lines 182-188: "The model results were evaluated according to (1) the flow regime visualized as flow lines (Fig. 2, Fig. 3); (2) the TDS distribution shown as snapshots at the end of the simulation (Fig. 3) as well as the standard deviation of the TDS concentration (SD) in each cell over the last 10a of simulation time period (Fig.. 4); and (3) the reaction potential (RP$_c$ = sum of accumulated mixing products in each cell (Mp$_C$) over the last 10a of simulation period) normalized to the absolute maximum Mp$_C$ concentration across all model versions (Fig. 5). The decision to assess SD TDS and RP$_c$ based on the final 10a of the simulation period was taken to circumvent the potential influence of the initial distribution of TDS, R$_s$ and R$_f$. Hence, the first 10a of the simulation period serve as model spin-up."

Technical Corrections

Line 60 led should be lead

AC: Was corrected as suggested.

Lines 67-69 This sentence does not make sense. Move citation to end of sentence.

AC: The sentence was modified as suggested:

Lines 67-70 "Greskowiak et al. (2023) concluded that redox zone dynamics in the STE are strongly affected by beach morphodynamics. While some redox reactions take place in the USPs and storm flood affected area, mixing controlled reactions driven by mixing of two solutes in different end members occur in the fringes of the USP and at the SW interface (Heiss et al., 2017)."

Table 1 Case 2 Description What does SF stand for? I think storm flood but this is not defined anywhere.

AC: We added the acronym for storm floods (SF) to the text.

Line 173-174: "Furthermore, cases with a stable beach morphology with and without storm floods (SF) as well as with three different permeability distributions (Fig. 1, c-f) were tested."

Line 261 reached should be reach and mowing should be moving

AC: We changed "reached" to "reach" and "mowing" to "moving".

Line 289 Where is the black cross?

AC: The reviewer is right there is no black cross. We removed it and modified the caption of Figure 6

Update caption Figure 6: "Figure 6: Cluster analysis (k-means,3) of all model cases based on model variation in salinity ($\gamma$) and the sum of reaction product of each model ($RP_M$). Individual model values were normalized to the base case (located at 1,1 in the plot)."

Figure 6 Some of the points are hard to see. It would be helpful if the scenarios were labeled so that each number could be read.

AC: We understand the need for a better recognition of model cases related to the cluster analysis. Reviewer #1 made a similar comment, so we decided that we will add the Cluster groups (A,B,C) to the respective subfigure in Figures 3 to 5 which are also mentioned in Table 1.

Line 308 lead should be led

AC: We changed it as suggested.

Line 322 Remove "related to"

AC: We removed it as suggested.

Line 331 Rephrase "impact significantly on the shape"

AC: We rephrased as suggested to:

Lines 360-362: "We found that in addition to morphodynamics and storm floods, the horizontal and vertical hydraulic conductivity, and longitudinal and transverse dispersivity significantly influence the shape, location, stability and extent of the USP, FDT and mixing controlled reactions."

Line 339 Rephrase "made that simulations ended"

AC: We rephrased the sentence to:

Lines 368-369: "Changing the values of hydraulic conductivity or dispersivity and their respective anisotropies caused the model cases to end up in different clusters."

References:

Greskowiak, J., Massmann, G., 2021. The impact of morphodynamics and storm floods on pore water flow and transport in the subterranean estuary. Hydrol. Process. 35, 1–5. https://doi.org/10.1002/hyp.14050

Greskowiak, J., Seibert, S.L., Post, V.E.A., Massmann, G., 2023. Redox-zoning in high-energy subterranean estuaries as a function of storm floods , temperatures , seasonal groundwater recharge and morphodynamics. Estuar. Coast. Shelf Sci. 290, 108418. https://doi.org/10.1016/j.ecss.2023.108418

Heiss, J.W., Post, V.E.A., Laattoe, T., Russoniello, C.J., Michael, H.A., 2017. Physical Controls on Biogeochemical Processes in Intertidal Zones of Beach Aquifers. Water Resour. Res. 53, 9225–9244. https://doi.org/10.1002/2017WR021110

---

## Editor Decision (ED1)

1. Line 16. Substitute "twenty-four" with "24".
2. Line 37. "The" with lower case after ":"-
3. Line 45. Substitute "small" with "fine". I know that "small/big scale" is often used in the scientific literature to denote small or big scale-lengths, but this is wrong, in my opinion. Think to geographical maps. A map at scale 1:1,000,000=10$^{-6}$ does not show many details: topographic maps at scale 1:10,000=10$^{-4}$ (i.e., 100 times greater!) provides many more details. Therefore, I prefer to use "fine/large scale".
4. Line 58. Substitute "effect" with "affect". "in the field of STE research" could be erased.
5. Line 70. Substitute "and analysed the development of redox zones. Greskowiak et al. (2023)" with ", analysed  the development of redox zones, and".
6. Line 89. Correct "boundary conditions and parameters a varied".
7. Line 90. Rephrase "a specific location. Specifically".
8. Line 93ff. Substitute "spec. stor." with a symbol, e.g., "$S_s$".
9. Line 105. Substitute "of 350-400 mm/a" either with "of about 350 mm" or with "varying between 350 mm/a and 400 mm/a". Similar modifications should be introduced in the rest of the paper, where ranges of values are mentioned. Please, follow the recommendation by NIST (https://www.nist.gov/pml/special-publication-811/nist-guide-si-check-list-reviewing-manuscripts), in particular those at point #7.
10. Lines 105 & 108. Substitute "approx.." with "approximately".
11. Line 125. Substitute "700 m long" with "700-meters-long". Substitute "of 2 m each" with "with a uniform horizontal length of 2 m" or something similar.
12. Lines 130 to 132. Such a flux corresponds to the Qf value defined at line 174, doesn't it? But Qf is not kept constant, it varies for some test cases, as shown in Table 1.
13. Lines 131 & 132. Substitute "specified flux of 0.5 m3/day per meter coastline" with "prescribed flux per unit coastline length of 0.5 m$^3$/(d m)". Correct the measurement units also in Table 1.
14. Line 139. Substitute "Feb-Jul" with "February to July XXXX", where XXXX should be replaced with the year in which the survey has been conducted.
15. Lines 139 & 140. Unify the format for "1m resolution", "six-month period", "20a simulation", and similar expression throughout the whole paper. I would prefer "one-meter resolution", "six-month-long period" or " a period of six months", "simulation for a period of 20 years".
16. Line 161. What is "PHT3D Eq. 1"? Probably, it is sufficient to erase "Eq. 1".
17. Line 163. Substitute ";" with ",".
18. Line 164. Add "," before "and". Word "formation" could be substituted with "production" or a synonymous.
19. Line 167. I would prefer "10$^{-7}$" instead of "1e-7". Analogous corrections could be done at line 170.
20. Lines 167 to 169. Rephrase the sentence "As Rf and Rs… from the value of k".
21. Line 171. Parentheses are needless.

22. Table 1. I do not understand the 9th column. If there are 3 storm floods per year, with 30 days between storm floods, does this mean that the storm flood has an average duration of about 92 days? In fact, (92 d + 30 d) x 3 = 366 d. Moreover, the description of storm flood modeling is missing, isn't it?

23. Lines 186 to 189. Expression "(RPc = model cases (Fig. 5)" is quite confusing, it should be rephrased.

24. Line 192. Erase "Eq. 2".

25. Line 194. Erase "Eq. 3".

26. Line 198. Erase "Eq. 4".

27. Figure 2, second line of the figure caption. Add "s" to "month". Substitute "3" with "three".

28. Line 223. Substitute "finer", possibly with "more finely".

29. Line 224. Is "but results otherwise" correct?

30. Line 244. Check "focused to".

31. Section 3. I am afraid that comparative adjective (e.g., higher, lower) are often used instead of superlative adjectives (e.g., highest, lowest). Please, check!

32. Lines 322 & 323. Rephrase sentence "Cluster A (red circles) had a γ (+/- 20%) and RPM (+/-20%) similar to the base case (located at the coordinates 1,1 in the plot in Fig. 6)", possibly as "Cluster A (red circles) is characterized by relatively small variations of γ and RPM with respect to the base case, namely variations in the range from -20 % to +20 %. In Figure 6, the base case corresponds to the point with coordinates (1,1)".

33. Line 324. Substitute "40-95 %" with "by more than 40 %". Substitute "30-70 %" with "by more than "30 %". See comment # 9.

34. Lines 327 & 328. Expression "was characterized by a lower γ, reduced by 40-80%, while keeping a RPM (+/-20%) similar to the base case" should be rephrased, possibly as "was characterized by values of γ reduced by more than 40 %, while RPM remains close to the base case (variations in the range from -20 % to +20 %)".

---

## Author Response (AR2)

Manuscript **hess-2024-196:** "*Effects of boundary conditions and aquifer parameters on salinity distribution and mixing-controlled reactions in high-energy beach aquifers.*"

Correspondence to Rena Meyer (rena.meyer@uni-oldenburg.de)

Author responses to Reviewer #1.

Dear Reviewer #1 we are happy that you are overall satisfied with the changes of our revised manuscript. We thank you for a second round of review and address your valuable remaining comments below point-by-point as indicated by Author Comments (AC) in blue, changes to the manuscript are indicated by "speech mark" and line numbers correspond to the clean version of the revised manuscript.

Kind regards, Rena Meyer

Reviewer Comments (RC)

1. As a suggestion, it might be worth adding a small sentence at the end of the abstract that describes the impact these findings have on coastal science in a broad sense.

AC: We extended the abstract and highlight the broader relevance of our findings for costal research:
" The present study advances the understanding of subsurface flow, transport and mixing processes that are dynamic beneath high-energy beaches. These processes control biogeochemical reactions that regulate nutrient fluxes to coastal ecosystems."

2. The authors added the sentence, "The decision to assess SD TDS and RPc based on the final 10a of the simulation period was taken in order to circumvent the potential influence of the initial distribution of TDS, Rs, and Rf." In line 185. Although this sentence addresses the possible influence of initial conditions, it is not clearly mentioned that the accumulated mixing concentration before the 10 years was subtracted from the final 20-year simulation concentration. I recommend adding this information as it might cause some confusion for the reader.

AC: We added the explanation how Mpc of the last 10a was calculated to the corresponding sentence in line 185:
"The model results were evaluated according to […] (3) the reaction potential ($RP_c$ = sum of the accumulated mixing products in each cell ($Mp_C$) over the last 10a of the simulation period, calculated by subtracting the accumulated mixing concentration of the first 10a from its final concentration of the 20a simulation) normalized to the absolute maximum $Mp_C$ concentration across all model cases (Fig. 5)."

3. The sentence in line 320: "Cluster B included the less dynamic and more stable cases with a lower γ,…" sounds redundant; consider rewording it.

AC: We rephrased the sentence to (l. 321):
" Cluster B (dark blue circles) showed a lower γ, reduced by 40-95%, and a lower RPM, reduced by 30-70%, compared to the base case. Cluster B contained the less dynamic and more

stable cases with either changing topography only (and no storm floods, case 21) or only storm floods (and no changing topography, case 2) or neither (case 1) and the low K case (case 5). Cluster C (light blue circles) was characterized by a lower $\gamma$, reduced by 40-80%, while keeping a RPM (+/- 20%) similar to the base case."

4. In the sentence in line 370: "… resulting in less variable salt patterns and less reactive conditions than the base case (cluster B) or less variable salt patterns but still reactive conditions compared to the base case (cluster C)." What do the authors mean by "still reactive conditions," consider removing the word "still."

AC: We removed "still" as suggested (l.371).

Dear Reviewer #2, we are happy that you are satisfied with our revised manuscript and accepted it.

---

## Author Response (AR3)

Manuscript **hess-2024-196**: "*Effects of boundary conditions and aquifer parameters on salinity distribution and mixing-controlled reactions in high-energy beach aquifers.*"

Correspondence to Rena Meyer (rena.meyer@uni-oldenburg.de)

Author responses to the Editor,

Dear Editor, we thank you for taking the time editing and reviewing our manuscript. We address your valuable comments below point-by-point as indicated by Author Comments (AC) in blue, changes to the manuscript are indicated by "speech mark" and line numbers correspond to the clean version of the revised manuscript.

Kind regards, Rena Meyer

Editor Comments (EC):

1. Line 16. Substitute "twenty-four" with "24".

AC: L.16 has been changed as suggested.

2. Line 37. "The" with lower case after ":"-

AC: L.37 has been changed as suggested.

3. Line 45. Substitute "small" with "fine". I know that "small/big scale" is often used in the scientific literature to denote small or big scale-lengths, but this is wrong, in my opinion. Think to geographical maps. A map at scale 1:1,000,000=10-6 does not show many details: topographic maps at scale 1:10,000=10-4 (i.e., 100 times greater!) provides many more details. Therefore, I prefer to use "fine/large scale".

AC: L.45 has been changed as suggested.

4. Line 58. Substitute "effect" with "affect". "in the field of STE research" could be erased.

AC: L.58 has been changed as suggested.

5. Line 70. Substitute "and analysed the development of redox zones. Greskowiak

et al. (2023)" with ", analysed the development of redox zones, and".

AC: L.70 has been changed as suggested.

6. Line 89. Correct "boundary conditions and parameters a varied".

AC: L.88 has been changed as suggested.

7. Line 90. Rephrase "a specific location. Specifically".

AC: Specifically has been removed.

8. Line 93ff. Substitute "spec. stor." with a symbol, e.g., "Ss".

AC: In the entire manuscript "spec. stor." has been substituted by $S_s$.

9. Line 105. Substitute "of 350-400 mm/a" either with "of about 350 mm" or with "varying between 350 mm/a and 400 mm/a". Similar modifications should be introduced in the rest of the paper, where ranges of values are mentioned. Please, follow the recommendation by NIST (https://www.nist.gov/pml/special-publication-811/nist-guide-si-check-list-reviewing-manuscripts), in particular those at point #7.

AC: L.106 has been changed as suggested. The whole manuscript has been checked for inconsistencies in ranges of values and the presentation has been updated according to the mentioned recommendations, throughout the manuscript.

10. Lines 105 & 108. Substitute "approx.." with "approximately".

AC: Lines 106 and 109 have been changed as suggested.

11. Line 125. Substitute "700 m long" with "700-meters-long". Substitute "of 2 m each" with "with a uniform horizontal length of 2 m" or something similar.

AC: L. 125 has been changed as suggested.

12. Lines 130 to 132. Such a flux corresponds to the Qf value defined at line 174, doesn't it? But Qf is not kept constant, it varies for some test cases, as shown in Table 1.

AC: We added ($Q_f$) to l. 130. Indeed, $Q_f$ is varied in the 24 model cases. As we describe in l. 112 the model set up is described based on the base case and the respective changes made in the 24 model cases are presented in Table 1. The value of the prescribed flux boundary in l. 130 to 132 hence refers to the base case.

13. Lines 131 & 132. Substitute "specified flux of 0.5 m3/day per meter coastline" with "prescribed flux per unit coastline length of 0.5 m3/(d m)". Correct the measurement units also in Table 1.

AC: L. 132 and Table 1 have been changed as suggested.

14. Line 139. Substitute "Feb-Jul" with "February to July XXXX", where XXXX should be replaced with the year in which the survey has been conducted.

AC: Lines 139 and 140 has been changed as suggested.

15. Lines 139 & 140. Unify the format for "1m resolution", "six-month period", "20a simulation", and similar expression throughout the whole paper. I would prefer "one-meter resolution", "six-month-long period" or " a period of six months", "simulation for a period of 20 years".

AC: Throughout the manuscript the format was changed as suggested.

16. Line 161. What is "PHT3D Eq. 1"? Probably, it is sufficient to erase "Eq. 1".

AC: Eq. 1 has been erased.

17. Line 163. Substitute ";" with ",".

AC: L. 163 has been changed as suggested.

18. Line 164. Add "," before "and". Word "formation" could be substituted with "production" or a synonymous.

AC: Line 164 has been changed as suggested.

19. Line 167. I would prefer "10-7" instead of "1e-7". Analogous corrections could be done at line 170.

AC: Lines 167 and 171 and Table 1 have been changed as suggested.

20. Lines 167 to 169. Rephrase the sentence "As Rf and Rs… from the value of k".

AC: We rephrased the sentence to (lines 168 to 171):

"As Rf and Rs were not removed by this processes, the relative differences of the mixing-controlled reaction potential between the different simulation cases are independent from the value of k. Thus the value of k has no further meaning, as long as it is greater than zero."

21. Line 171. Parentheses are needless.

AC: L. 171 has been changed as suggested.

22. Table 1. I do not understand the 9th column. If there are 3 storm floods per year, with 30 days between storm floods, does this mean that the storm flood has an average duration of about 92 days? In fact, (92 d + 30 d) x 3 = 366 d. Moreover, the description of storm flood modeling is missing, isn't it?

AC: The duration of each storm flood is 1 day. Usually few storm floods may occur from mid-September to mid-April. Three storm floods with each 30 days in between means that there are three storm floods within the winter season (at day 1, day 31 and day 62). In lines 147-150 the modelling of the storm floods is described. We extended the description by (l. 104):

"The northern beach can be affected by storm floods that reach up to the base of the dunes from mid-September to mid-April, with storm floods most likely to occur in the winter months and lasting for one or two days."

23. Lines 186 to 189. Expression "(RPc = model cases (Fig. 5)" is quite confusing, it should be rephrased.

AC: we rephrased the section to (l. 188-196):

"The model results were evaluated according to (1) the flow regime visualized as flow lines (Fig. 2, Fig. 3); (2) the TDS distribution shown as snapshots at the end of the simulation (Fig. 3), as well as the standard deviation of the TDS concentration (SD) in each cell over the last 10 years of the simulation period (Fig. 4); and (3) the reaction potential $RP_c$ (Eq. 2) normalized to the absolute maximum $Mp_C$ concentration across all model cases (Fig. 5). Here, RPc is the sum of the accumulated mixing products in each cell ($Mp_C$) over the last 10 years of the simulation period, calculated by subtracting the accumulated mixing concentration of the first 10 years from the final concentration at the end of the 20 year simulation period .The decision to evaluate SD TDS and $RP_c$ based on the last 10 years of the simulation period was taken to avoid the potential influence of the initial distribution of TDS, Rs and Rf. Therefore, the first 10 years of the simulation period serve as a model spin-up."

24. Line 192. Erase "Eq. 2".

AC: L.193 has been changed as suggested.

25. Line 194. Erase "Eq. 3".

AC: : L.195 has been changed as suggested.

26. Line 198. Erase "Eq. 4".

AC: : L.200 has been changed as suggested.

27. Figure 2, second line of the figure caption. Add "s" to "month". Substitute "3" with "three".

AC: : The caption of figure 2 has been changed as suggested.

28. Line 223. Substitute "finer", possibly with "more finely".

AC: L. 225 has been changed as suggested.

29. Line 224. Is "but results otherwise" correct?

AC: L. 226 "otherwise" has been removed .

30. Line 244. Check "focused to".

AC: The sentence in l. 245 has been changed to:

"The high RP zone was concentrated in the area affected by the  storm floods, the deeper USP and the wedge interface (Fig. 5, case 6)."

31. Section 3. I am afraid that comparative adjective (e.g., higher, lower) are often used instead of superlative adjectives (e.g., highest, lowest). Please, check!

AC: We are not sure if we understand this comment correctly. In section 3 we present the results of the 24 simulation cases and compare the cases to the base case. Therefore we use the comparative adjectives. We are not aware of what should be changed here. We added (l.210):

"The results of the different model variants were compared to the base case."

32. Lines 322 & 323. Rephrase sentence "Cluster A (red circles) had a γ (+/- 20%) and RPM (+/-20%) similar to the base case (located at the coordinates 1,1 in the plot in Fig. 6)", possibly as "Cluster A (red circles) is characterized by relatively small variations of γ and RPM with respect to the base case, namely variations in the range from -20 % to +20 %. In Figure 6, the base case corresponds to the point with coordinates (1,1)".

AC: Lines 324 to 325 have been changed as suggested.

33. Line 324. Substitute "40-95 %" with "by more than 40 %". Substitute "30-70 %" with "by more than "30 %". See comment # 9.

AC: L. 326 has been changed as suggested.

34. Lines 327 & 328. Expression "was characterized by a lower γ, reduced by 40-80%, while keeping a RPM (+/-20%) similar to the base case" should be rephrased, possibly as "was characterized by values of γ reduced by more than 40 %, while RPM remains close to the base case (variations in the range from -20 % to +20 %)".

AC: Lines 327 and 328 have been changed as suggested.